# AKAP79 enables calcineurin to directly suppress protein kinase A activity

Timothy W Church[1], Parul Tewatia[2,3], Saad Hannan[1], João Antunes[2], Olivia Eriksson[2], Trevor G Smart[1], Jeanette Hellgren Kotaleski[2,3], Matthew G Gold[1]*

[1]Department of Neuroscience, Physiology & Pharmacology, University College London, London, United Kingdom; [2]Science for Life Laboratory, School of Electrical Engineering and Computer Science, KTH Royal Institute of Technology, Stockholm, Sweden; [3]Department of Neuroscience, Karolinska Institute, Stockholm, Sweden

**Abstract** Interplay between the second messengers cAMP and $Ca^{2+}$ is a hallmark of dynamic cellular processes. A common motif is the opposition of the $Ca^{2+}$-sensitive phosphatase calcineurin and the major cAMP receptor, protein kinase A (PKA). Calcineurin dephosphorylates sites primed by PKA to bring about changes including synaptic long-term depression (LTD). AKAP79 supports signaling of this type by anchoring PKA and calcineurin in tandem. In this study, we discovered that AKAP79 increases the rate of calcineurin dephosphorylation of type II PKA regulatory subunits by an order of magnitude. Fluorescent PKA activity reporter assays, supported by kinetic modeling, show how AKAP79-enhanced calcineurin activity enables suppression of PKA without altering cAMP levels by increasing PKA catalytic subunit capture rate. Experiments with hippocampal neurons indicate that this mechanism contributes toward LTD. This non-canonical mode of PKA regulation may underlie many other cellular processes.

*For correspondence:
m.gold@ucl.ac.uk

Competing interest: The authors declare that no competing interests exist.

## Introduction

Cyclic adenosine monophosphate (cAMP) and $Ca^{2+}$ are ancient second messengers that are fundamental to the regulation of many dynamic cellular processes including synaptic plasticity (*Huang et al., 1994*), heart contraction (*Bers et al., 2019*), and glycogen metabolism (*Roach et al., 2012*). Crosstalk between the two second messengers is a common feature of cellular signaling. For example, cAMP can enhance cytosolic $Ca^{2+}$ entry by triggering phosphorylation of key ion channels (*Qian et al., 2017*; *Schmitt et al., 2003*) by its major intracellular receptor cAMP-dependent protein kinase, also known as Protein Kinase A (PKA). Similarly, $Ca^{2+}$ can regulate cAMP levels by altering activities of both phosphodiesterases (*Baillie et al., 2019*) and adenylyl cyclases (*Qi et al., 2019*). At the receptor level, a common signaling motif is the opposition of PKA and the highly abundant $Ca^{2+}$-sensitive phosphatase calcineurin (CN), with CN triggering cellular changes by removing phosphate from substrates primed by PKA. Notable examples of this motif are the regulation of postsynaptic substrates including AMPA-type glutamate receptors in the induction of long-term depression (LTD) of synaptic strength (*Bear, 2003*), and control of NFAT nuclear localization in immune responses (*Hogan, 2017*). According to current consensus, in these cases CN dephosphorylates substrates without any requirement for directly altering PKA activity (*Dittmer et al., 2014*; *Lu et al., 2011*; *Tunquist et al., 2008*; *Weisenhaus et al., 2010*; *Zhang and Shapiro, 2016*). This implies that energetically costly futile cycles of phosphate addition and removal by PKA and CN must persist to maintain dephosphorylated substrate. It would be more logical for PKA activity to be switched off when CN is activated during substrate dephosphorylation. Uncovering the mechanism to achieve this is the focus of this study.

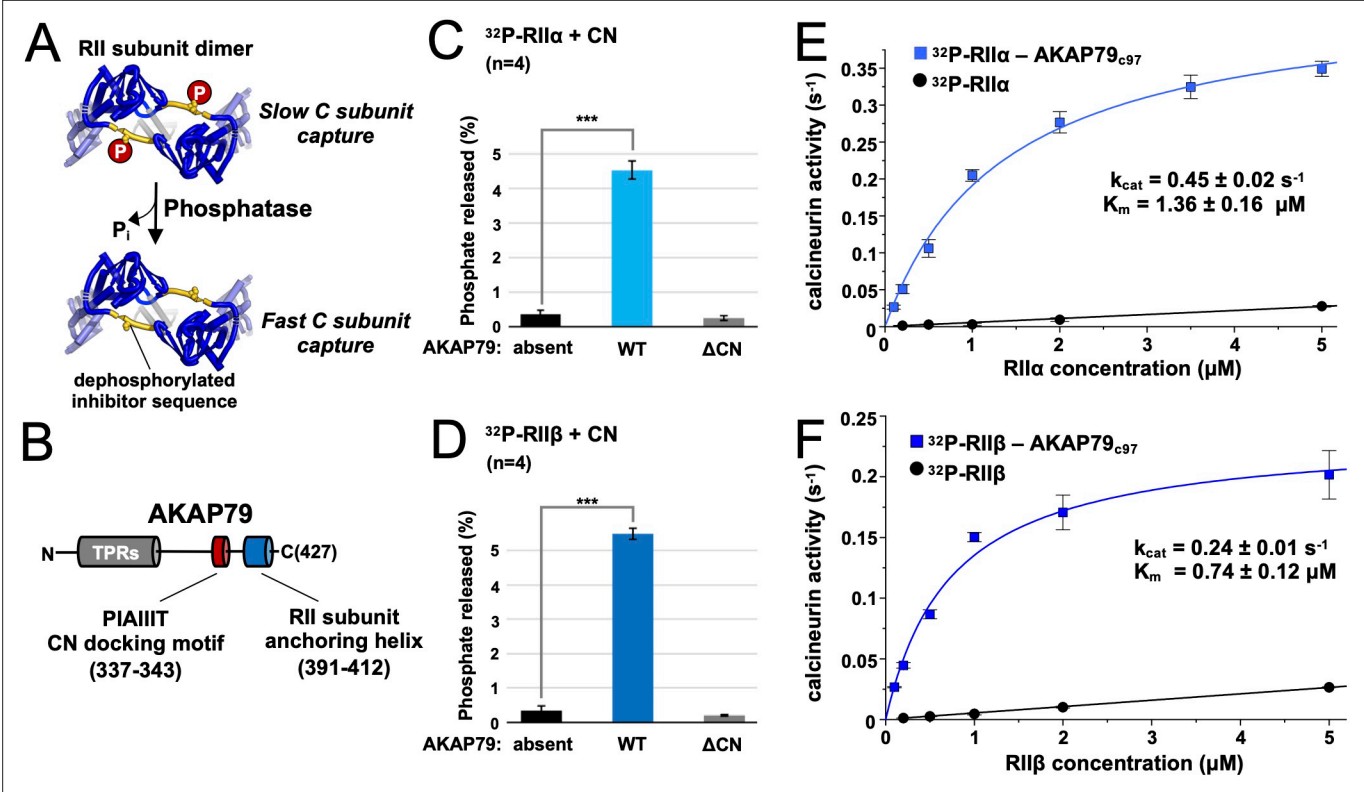

**Figure 1.** Effect of AKAP79 on pRII dephosphorylation by CN. (**A**) Dephosphorylation of the inhibitor sequence (IS, yellow) of RII subunits enables faster PKA C subunit capture. (**B**) AKAP79 contains anchoring sites for CN (red) and PKA RII subunits (blue) in its C-terminal region. Other macromolecular interactions are mediated through elements within its tandem polybasic regions (TPRs, gray). (**C**) CN-catalyzed phosphate release from pRIIα subunits with either no AKAP79, WT AKAP79 (light blue), or AKAP79 lacking the PIAIIIT anchoring motif ('ΔCN'). (**D**) CN-catalyzed phosphate release from pRII$\beta$ subunits with either no AKAP79, WT AKAP79 (dark blue), or AKAP79 ΔCN. (**E**) The relationship between CN activity toward pRIIα subunits and pRIIα concentration with pRIIα subunits included either alone (black circles) or in complex with AKAP79$_{c97}$ (light blue squares). (**F**) The relationship between CN activity towards pRII$\beta$ subunits and pRII$\beta$ concentration with pRII$\beta$ subunits included either alone (black circles) or in complex with AKAP79$_{c97}$ (dark blue squares). For panels E and F, activities at each concentration were measured in triplicate. Statistical comparisons were performed using two-tailed unpaired Student $t$-tests. ***$p < 0.001$.

The online version of this article includes the following figure supplement(s) for figure 1:

**Source data 1.** Radioactive phosphatase assays.

**Figure supplement 1.** Purified proteins.

**Figure supplement 2.** pRII phosphorylation by CN at supra-physiological concentrations.

**Figure supplement 2—source data 1.** Phosphatase assays without AKAP79.

**Figure supplement 3.** Effect of AKAP79c97 variants on pNPP and pRII phosphopeptide dephosphorylation.

**Figure supplement 3—source data 1.** Colorimetric phosphatase assays.

Recent years have seen renewed interest in mechanisms for regulating the release and re-capture of PKA catalytic subunits (*Bock et al., 2020*; *Gold, 2019*; *Zhang et al., 2020*), including new data that hint at how CN might directly suppress PKA activity. PKA is comprised of regulatory subunit dimers that bind and sequester PKA catalytic (C) subunits in an inhibited state (*Taylor et al., 2019*). PKA regulatory subunits are classified into type I subunits (RIα and RIβ) that are predominantly cytosolic, and type II subunits (RIIα and RIIβ) that co-sediment with membranes (*Gold, 2019*). The regulatory subunit inhibitor sequence (IS) is phosphorylated upon association with C subunits for RII but not RI subunits, which bear alanine in place of serine in the phospho-acceptor site (S98 in RIIα). Quantitative immunoblotting and mass spectrometry (MS) have recently shown that PKA regulatory subunits – and particularly RII subunits – greatly outnumber PKA C subunits (*Aye et al., 2010*; *Walker-Gray et al., 2017*) throughout the body. In addition, *Zhang et al., 2015* have extended earlier observations (*Rangel-Aldao and Rosen, 1976*) to quantify differences in the rate of C subunit binding to RII subunits either

phosphorylated (pRII) or dephosphorylated at the IS. Remarkably, the $k_{on}$ rate for C subunit association is ~50 times faster for dephosphorylated RII than pRII (*Zhang et al., 2015*; *Figure 1A*). In theory, rapid dephosphorylation of RII subunits by CN could directly suppress PKA activity by increasing the rate of C subunit capture thereby reducing the proportion of C subunits that are dissociated (*Buxbaum and Dudai, 1989*; *Isensee et al., 2018*; *Ogreid and Døskeland, 1981*; *Stemmer and Klee, 1994*; *Zhang et al., 2015*; *Zhang et al., 2012*). While recent observations concerning PKA subunit stoichiometry and pRII/RII binding kinetics support this notion, isolated pRII is a low-affinity substrate for CN with a half-maximal substrate concentration ($K_m$) above 20 µM (*Blumenthal et al., 1986*; *Perrino et al., 1992*; *Stemmer and Klee, 1994*). Therefore, pRII dephosphorylation by CN would not be expected to occur to a meaningful degree at physiological concentrations in the absence of an additional factor.

Anchoring proteins support signal transduction by elevating effective local concentrations of signaling proteins, and therefore theoretically an AKAP might support pRII dephosphorylation by CN in cells (*Gildart et al., 2020*). A-kinase anchoring protein 79 (AKAP79; rodent ortholog AKAP150, gene name AKAP5) is a prototypical mammalian anchoring protein with several features that indicate it could operate in part by increasing the effective protein concentration of pRII subunits for CN. AKAP79 can simultaneously anchor both CN and PKA (*Coghlan et al., 1995*). It contains an amphipathic anchoring helix (*Gold et al., 2006*; *Kinderman et al., 2006*) for binding RII subunits, and a short linear 'PIAIIIT' CN anchoring motif (*Dell'Acqua et al., 2002*; *Li et al., 2012*). The two anchoring sites are separated by only ~50 amino acids in the primary sequence within the C-terminus of AKAP79 (*Figure 1B*). AKAP79 is localized in dendritic spines where it is required for anchoring RII subunits (*Tunquist et al., 2008*). The anchoring protein is necessary for both induction of long-term depression (LTD) of CA3-CA1 hippocampal synapses (*Lu et al., 2008*; *Tunquist et al., 2008*; *Weisenhaus et al., 2010*), and for CN-mediated dephosphorylation of NFAT (*Kar et al., 2014*; *Murphy et al., 2014*) – both processes that are driven by CN dephosphorylation of sites primed by PKA. Despite these characteristics, the possibility that AKAP79 could support pRII dephosphorylation by CN has been disregarded perhaps because paradoxically AKAP79 acts as a weak inhibitor for CN dephosphorylation of 20-mer peptides corresponding to the phosphorylated RII IS (*Coghlan et al., 1995*; *Kashishian et al., 1998*). We reasoned that these assays could be misleading since peptide substrates are not subject to anchoring alongside CN that occurs for full-length RII subunits. To resolve this issue, in this study we measured how AKAP79 alters CN activity towards full-length pRII subunits. We went on to determine if AKAP79 can reduce the fraction of dissociated C subunits in concert with CN using fluorescence-based assays supported by kinetic modeling, before substantiating our observations in hippocampal neurons.

## Results

### AKAP79 enables CN to efficiently dephosphorylate RII subunits at physiological concentrations

We set out to determine whether AKAP79 can increase CN dephosphorylation of full-length RII subunits phosphorylated at the IS. Using purified proteins (*Figure 1—figure supplement 1*), we compared $^{32}$P release from either pRIIα (*Figure 1C*) or pRIIβ (*Figure 1D*). Thirty second reactions were initiated by addition of excess $Ca^{2+}$/calmodulin (CaM) to 10 nM CN and 400 nM pRII subunits. For pRIIα without AKAP79, phosphate was released from only 0.36% ± 0.13 % of the subunits (black, *Figure 1C*). Inclusion of full-length AKAP79 in the reaction mix increased phosphate release by 12.4-fold ($P = 7.4 \times 10^{-6}$) to 4.52% ± 0.26 % pRIIα subunits (light blue, *Figure 1C*). Removing the PIAIIIT anchoring sequence in AKAP79 (ΔCN) returned dephosphorylation to a baseline level of 0.26% ± 0.06 % (gray, *Figure 1C*), consistent with a mechanism in which anchoring of CN adjacent to pRII subunits enhances the rate of dephosphorylation. Similar results were obtained for pRIIβ, with addition of AKAP79 increasing phosphate release 16.3-fold (p = $3.0 \times 10^{-6}$) from 0.34% ± 0.13 % (black, *Figure 1D*) to 5.49% ± 0.17 % (dark blue, *Figure 1D*). Ablating the CN anchoring site in AKAP79 also reduced phosphorylation to a baseline level of 0.2% ± 0.02 % for this isoform (gray, *Figure 1D*).

We next measured CN activity toward pRII over a range of pRII concentrations. We compared activity towards pRII subunits alone or in complex with a fragment of AKAP79 (AKAP79$_{c97}$) encompassing positions 331–427 that includes the CN and RII subunit anchoring sites. Working with this stable highly-expressed construct enabled us to purify sufficient quantities of pRIIα-AKAP79$_{c97}$ and

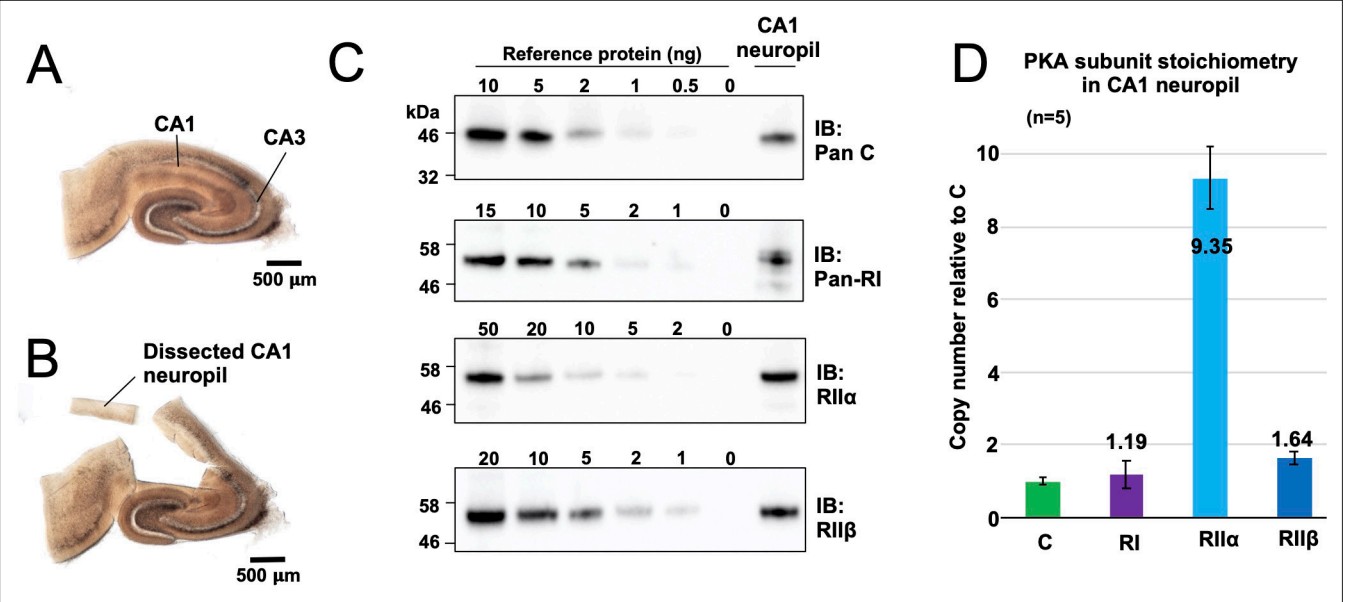

**Figure 2.** Quantitation of PKA subunits in CA1 neuropil. Images of a P17 rat hippocampal slice before (**A**) and after (**B**) micro-dissection of the CA1 neuropil layer. (**C**) Immunoblots of CA1 neuropil extract for PKA subunits. Extracts were run alongside reference amounts of the relevant purified PKA subunit in each immunoblot (**Figure 2—figure supplement 1**). In each case, 15 µg total protein extract was run alongside the reference series, with the exception of the anti-C immunoblot (10 µg extract). (**D**) Copy numbers of PKA subunits in rat CA1 neuropil normalized to C subunits.

The online version of this article includes the following figure supplement(s) for figure 2:

**Source data 1.** Quantitative immunoblotting.

**Figure supplement 1.** Reference curves for quantitation of PKA subunits in CA1 neuropil.

pRIIβ-AKAP79$_{c97}$ complexes (**Figure 1—figure supplement 1C & D**) to sample concentrations up to 5 µM. In complex with AKAP79$_{c97}$, both pRIIα and pRIIβ acted as relatively high affinity substrates of CN. pRIIα-AKAP79$_{c97}$ (light blue, **Figure 1E**) was dephosphorylated with a half-maximal concentration ($K_m$) of 1.36 ± 0.16 µM and turnover number ($k_{cat}$) of 0.45 ± 0.02 s$^{-1}$, and pRIIβ-AKAP79$_{c97}$ with $K_m$ = 0.74 ± 0.12 µM and $k_{cat}$ = 0.24 ± 0.01 s$^{-1}$. As expected, in the absence of the anchoring protein, pRIIα and pRIIβ subunits served as low affinity substrates for CN (black lines, **Figure 1E & F**). For both isolated pRII isoforms, the relationship between phosphatase activity and pRII concentration was linear up to 20 µM (**Figure 1—figure supplement 2**) – the highest concentration tested – indicative of a $K_m$ of greater than 20 µM. CN activity was very low ( < 0.03 s$^{-1}$) at concentrations of 5 µM pRII or lower. This is consistent with earlier studies that reported a $K_m$ of 94 µM for CN dephosphorylation of a phosphorylated 19-mer peptide derived from the RIIα IS (**Stemmer and Klee, 1994**). We also compared CN activity toward para-nitrophenylphosphate (pNPP) and a peptide corresponding to the isolated phosphorylated RII inhibitor sequence (sequence DLDVPIPGRFDRRVpSVAAE) with and without variants of AKAP79$_{c97}$ (**Figure 1—figure supplement 3**). WT AKAP79$_{c97}$ enhanced CN activity toward pNPP by ~65%, and reduced its activity towards pRII phosphopeptide by ~50%, consistent with previous reports that AKAP79 acts as a weak inhibitor of CN activity toward this phosphopeptide (**Coghlan et al., 1995**; **Kashishian et al., 1998**). Enhanced CN activity toward pNPP in the presence of PxIxIT-type motifs that have the opposite effect on phosphopeptide dephosphorylation has also been observed previously (**Grigoriu et al., 2013**). Overall, our data are consistent with a mechanism in which AKAP79 enhances CN dephosphorylation of full-length RII subunits by increasing effective substrate concentration rather than by altering the inherent catalytic activity of the phosphatase.

To put our kinetic parameters for pRII dephosphorylation into a physiological context, we set out to determine accurate protein concentrations for PKA subunits in the CA1 neuropil where Schaffer collaterals from the CA3 region synapse onto CA1 dendrites (**Figure 2A**). These synapses are a leading prototype for understanding forms of LTD driven by CN following PKA priming (**Bear, 2003**). We collected hippocampal slices from 18 -day-old male Sprague-Dawley rats before micro-dissecting CA1 neuropil sections (**Figure 2B**). Following homogenization, concentrations of C, RIIα, RIIβ, and

RI subunits in the extracted protein were determined using quantitative immunoblotting by running extracts (n = 5) alongside reference concentrations of purified PKA subunits (*Figure 2C*, *Figure 2— figure supplement 1*; *Walker-Gray et al., 2017*). We found that RIIα was by far the most predominant PKA subunit in the CA1 neuropil, accounting for 0.32% ± 0.029 % total protein content compared to 0.032% ± 0.003 % for C subunits, 0.041% ± 0.014 % for RI, and 0.06% ± 0.006 % for RIIβ. These numbers equate to a 9.4-fold higher molar abundance of RIIα subunits (light blue, *Figure 2D*) relative to C subunits with RI and RIIβ present at similar levels to C subunits. The predominance of the RIIα subunit is consistent with a previous imaging study of rodent hippocampus (*Weisenhaus et al., 2010*). Assuming that protein accounts for 8 % of total rat brain weight (*Clouet and Gaitonde, 1956*), we estimated RII subunit concentrations of 5.9 µM (RIIα) and 1.03 µM (RIIβ). These values fall within the range where CN efficiently dephosphorylates pRII only in the presence of AKAP79 (*Figure 1E & F*). Taken together our data therefore indicate that AKAP79 greatly enhances CN activity toward phosphorylated RII subunits at physiological concentrations.

## AKAP79 enables calcineurin to suppress type II PKA activity

Given that AKAP79 supports rapid pRII dephosphorylation by CN, we hypothesized that the AKAP could enable CN to directly reduce the fraction of dissociated C subunits in mixtures of RII and C subunits. To test this hypothesis, we utilized purified A-kinase activity reporter 4 (AKAR4) (*Figure 3A*). PKA phosphorylation at threonine within the reporter's central 'LRRA**T**LVD' motif leads to a conformational change that increases FRET efficiency between the terminal fluorescent proteins (*Figure 3A*; *Depry et al., 2011*). All AKAR4 experiments were performed using purified protein mixtures in 96-well plates. For each recording, three baseline 520/485 nm emission ratios were measured prior to injection of ATP and the desired concentration of cAMP into the protein mixture to initiate phosphorylation. Emission ratios were collected once every 5 s thereafter. In calibration experiments with AKAR4 and different concentrations of C subunit only (*Figure 3—figure supplement 1A*), we found that the initial rate of AKAR4 phosphorylation had a close to linear relationship to C subunit concentration up to 400 nM C subunit (*Figure 3—figure supplement 1B*). Full AKAR4 phosphorylation increased the emission ratio by 72 % (*Figure 3—figure supplement 1A*), consistent with previous studies (*Depry et al., 2011*). Importantly, supplementing these reactions with 1.5 µM activated CN had no detectable effect on AKAR4 phosphorylation rates, indicating that the phosphatase cannot efficiently dephosphorylate the reporter (*Figure 3—figure supplement 1C & D*). In comparison, supplementation with 1.5 µM protein phosphatase 1 (PP1) reduced the phosphorylation rate by ~7- fold (*Figure 3—figure supplement 2A & B*). Phosphatase assays using pre-phosphorylated AKAR4 confirmed that CN exhibits very limited activity towards the reporter (*Figure 3—figure supplement 2C & D*), such that the reporter is well suited for experiments focusing on direct suppression of PKA activity by CN.

Next, we assembled purified protein mixtures with the aim of mimicking signaling involving PKA, CN, and AKAP79 in CA1 dendritic spines. RIIα, RIIβ, and C subunits were included at concentrations determined in CA1 neuropil extracts (*Figure 2*). CaM was added at a molar excess of 5 µM, CN at 1.5 µM (*Goto et al., 1986*), and AKAP79$_{c97}$ – when included – at half the concentration of total RII subunits (summarized in *Figure 3B*). RI subunits were omitted since they are not thought to be present in dendritic spines (*Ilouz et al., 2017*; *Tunquist et al., 2008*), and because the RI inhibitor site is not phosphorylated so cannot be regulated by CN. We first monitored AKAR4 phosphorylation in reactions containing RIIα, RIIβ, C, and CaM (black, *Figure 3C*). Increasing the concentration of cAMP injected alongside ATP raised rates of AKAR4 phosphorylation as expected (black bars, *Figure 3D*). Supplementing the reactions with CN led to small but consistent decreases in the rate of AKAR4 phosphorylation at all cAMP concentrations (blue, *Figure 3C & D*). Rates were determined between 30 and 90 s in the linear early phase that followed a brief ~15 second delay, with the exception of the lowest two cAMP concentrations (0 and 100 nM), where relatively slow rates were calculated between 30 and 330 s. Additional supplementation with AKAP79$_{c97}$ markedly decreased the rate of AKAR4 phosphorylation (red, *Figure 3C*). For example, with 1 µM cAMP activation, addition of both CN and the AKAP reduced the initial rate of AKAR4 phosphorylation by 2.8-fold from 18.9 ± 0.6 to 6.7% ± 0.8% per minute (p = 0.0007, black and red bars, *Figure 3D*). To confirm that AKAP79 enables CN to suppress PKA activity by anchoring it alongside RII subunits, we investigated the effect of removing either the CN (positions 337–343) or PKA (391-400) anchoring sites. At 1 µM cAMP activation, addition of wild-type (WT) AKAP79$_{c97}$ (red, *Figure 3E & F*) reduced the initial rate of AKAR4 phosphorylation by

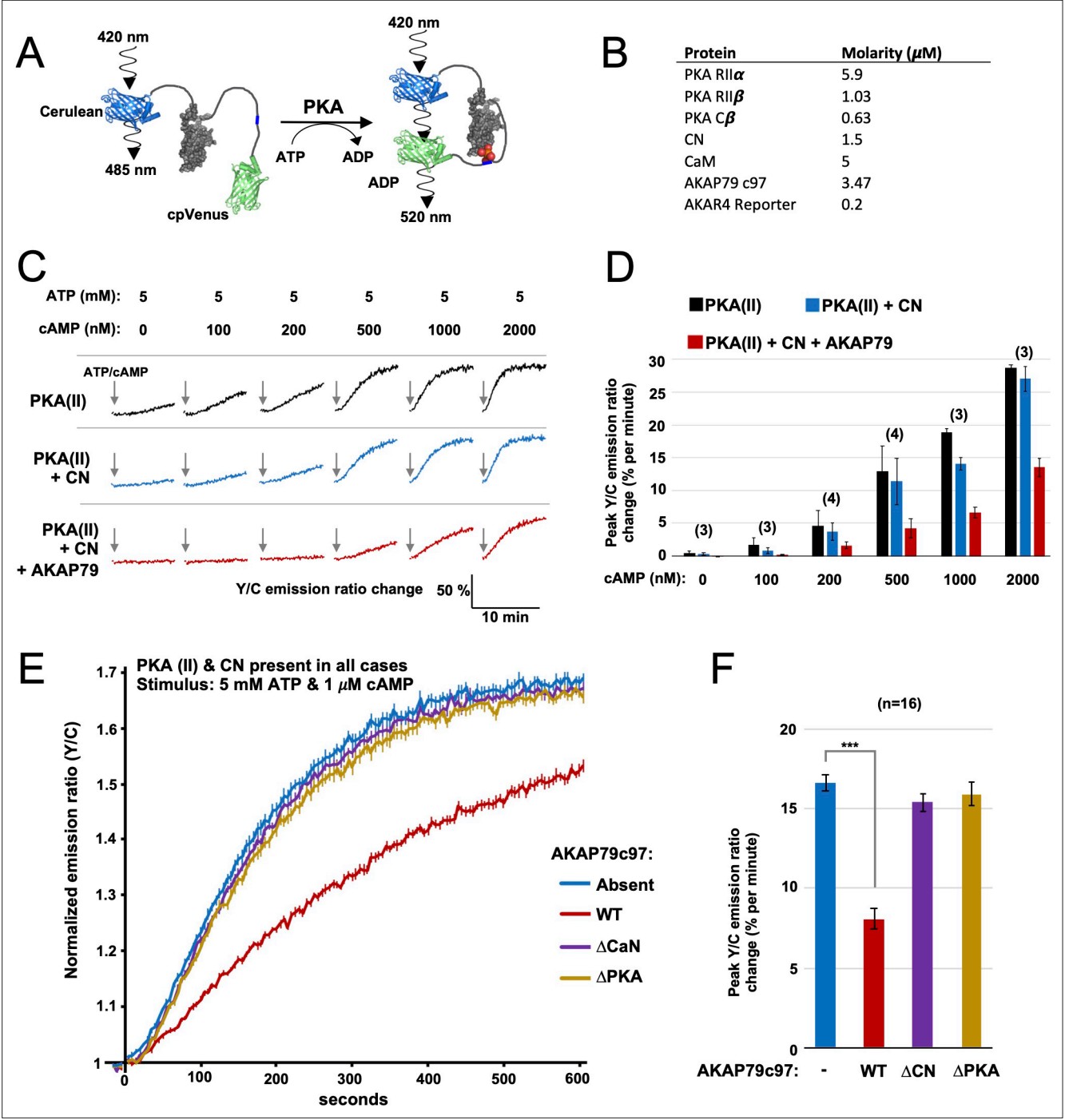

**Figure 3.** FRET-based PKA activity measurements. (**A**) AKAR4 mechanism: phosphorylation of the sensor by PKA is detected as an increase in FRET between the terminal fluorescent proteins. (**B**) Concentrations of proteins used for in vitro AKAR4 assays. Different experiments utilized different mixtures of these proteins but always at these concentrations. (**C**) Representative AKAR4 traces showing change in 520 nm / 485 nm (Y/C) emission ratio over time after injection of different concentrations of cAMP in tandem with 5 mM ATP. All protein mixtures included AKAR4, type II PKA (RIIα, RIIβ, C), and CaM. Experiments were performed with either no further additives (top row, black), with CN added (middle row, blue), or with both CN and AKAP79$_{c97}$ added (bottom row, red). ATP/cAMP injections are indicated by arrows. (**D**) The chart shows peak rates of emission ratio change for the recordings shown in the preceding panel. n values are stated above the columns. (**E**) For these recordings, type II PKA, CN, and CaM were included in all cases. Phosphorylation was initiated by injection of 5 mM ATP and 1 μM cAMP at t = 0. Averaged responses ± standard error (SE) are shown with no further additives (blue), or when either WT (red), ΔCN (purple), or ΔPKA (gold) variants of AKAP79$_{c97}$ were included. (**F**) Peak rates (calculated between 30 and 90 s) for the responses shown in the preceding panel. Statistical comparisons were performed using two-tailed unpaired Student $t$-tests. ***p < 0.001.

*Figure 3 continued on next page*

*Figure 3 continued*

The online version of this article includes the following figure supplement(s) for figure 3:

**Source data 1.** Rates of AKAR4 phosphorylation in purified protein mixtures.

**Figure supplement 1.** AKAR4 reference measurements with PKA catalytic subunit.

**Figure supplement 1—source data 1.** Rates of AKAR4 phosphorylation with C subunit alone.

**Figure supplement 2.** Comparison of CN and PP1 activity towards AKAR4.

**Figure supplement 2—source data 1.** Comparison of CN and PP1 activity toward pAKAR4.

---

2.06-fold (p = 2.7 x 10$^{-11}$) compared to supplementation with only CN (blue). Similar AKAR4 responses were obtained when either the AKAP was omitted altogether (blue, *Figure 3E & F*), or if either the CN (purple) or PKA (orange) anchoring sites in the AKAP were removed. Overall, these AKAR4 measurements reveal that AKAP79 enables CN to robustly decrease type II PKA activity by anchoring the two enzymes together.

## Mechanistic basis of PKA suppression by calcineurin and AKAP79

We next aimed to quantify how AKAP79 and CN changed the fraction of free C subunits in our reaction mixtures. To estimate this, we cross-referenced rates of AKAR4 phosphorylation recorded in the 'spine mimic' reaction mixtures (*Figure 3C & E*) to the reference curve (r = 0.998) obtained with only C subunits (*Figure 3—figure supplement 1B*). We focused on determining free C subunit concentrations during the early period of linear change (30–90 s for cAMP concentrations of 0.2 μM and above) where we assume the underlying kinetics are close to equilibrium. We calculated free C subunit concentrations following this approach using all available data between 0 and 2 μM cAMP (*Figure 3—figure supplement 1E*). The calculated proportion of C subunits that are dissociated at different cAMP concentrations are shown for type II PKA+ CaM either alone (black, *Figure 4A*), with CN (blue, *Figure 4B*), or with both CN and AKAP79$_{c97}$ (red, *Figure 4C*). Together, AKAP79 and CN reduced the proportion of free C subunits at equilibrium across the cAMP concentration range including from 47.8 ± 1.5 to 20.2% ± 0.8% at 1 μM cAMP, and from 65.7 ± 1.1 to 33.2% ± 3.3% at 2 μM cAMP (*Figure 4A & C*). The effect of adding CN alone was limited (*Figure 4B*), consistent with the much lower activity of the phosphatase towards pRII subunits in the low micromolar range (*Figure 1E & F*).

To understand at a deeper level how CN and AKAP79 reduce the fraction of free C subunits, we updated and extended a kinetic model (*Buxbaum and Dudai, 1989*) that takes into account transitions between pRII (left-hand square, *Figure 4D*) and unphosphorylated RII subunits (right-hand square). The extended model also incorporates AKAR4 binding to and phosphorylation by free C subunits. We used a Bayesian approach (*Eriksson et al., 2019*) to estimate parameter sets for the model that could fit data pooled from AKAR4 recordings obtained after stimulation with 0, 0.2, 1, and 2 μM cAMP (*Figure 3C & E*). A log uniform prior parameter distribution was used as a starting point for the Bayesian method, where each parameter was allowed to vary three orders of magnitude around a default value (*Supplementary file 1*). The default values were taken from empirical data, including rates of pRII dephosphorylation determined in this study (*Figure 1*), and binding rates of C subunits to pRII and RII (*Zhang et al., 2015*). This parameter estimation approach resulted in approximately 15,000 parameter sets that could explain the experimental data (*Figure 4E–G*). Simulations using these parameter sets enabled us to predict concentration changes of individual states within the model that cannot be determined experimentally (first three columns, *Figure 4—figure supplement 1*). The model indicates that AKAP79 and CN together shift C subunit capture to the faster right-hand square sub-system (*Figure 4D*), driving down the fraction of free C subunits and thereby reducing PKA activity.

## Mutation of the RIIα IS phosphorylation site occludes PKA suppression by CN

The results of the preceding sections show that AKAP79 targeting of CN for direct suppression of PKA is a viable mechanism for LTD induction. Previously published studies in hippocampal slices involving genetic manipulation of AKAP150 (the rodent ortholog of AKAP79) are also consistent with this mechanism. Full AKAP150 knock-out (*Lu et al., 2008*; *Tunquist et al., 2008*; *Weisenhaus et al., 2010*), or AKAP150 knock-in with variants lacking either the PKA or CN anchoring sites (*Jurado et al., 2010*;

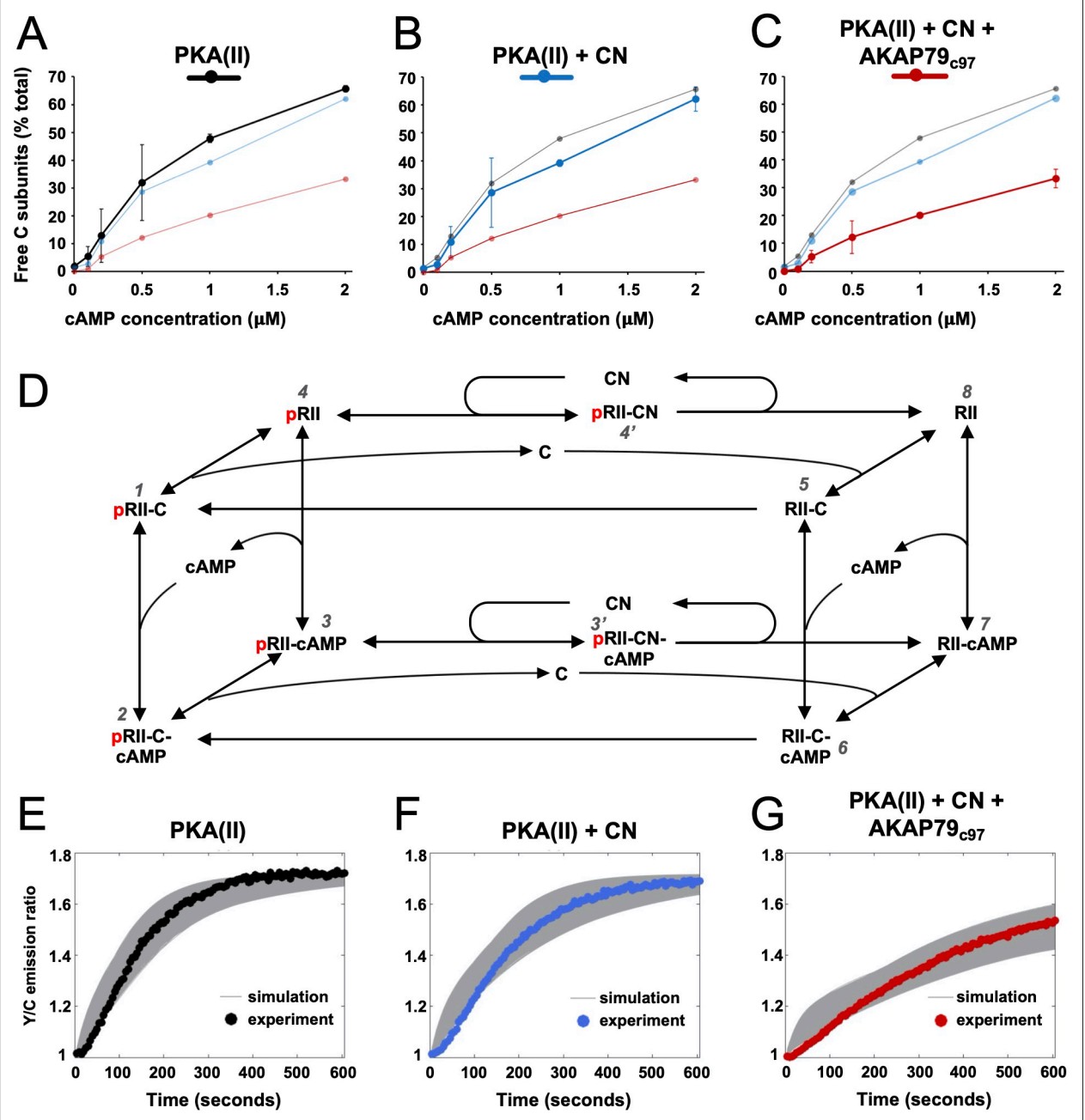

**Figure 4.** Kinetic analysis of PKA-CN-AKAP79 signaling. (**A–C**) Estimates of the average proportion of free C subunits between 30 and 90 s for type II PKA alone (black), with CN (blue), and with both CN and AKAP79$_{c97}$ (red) following activation of the protein mixtures with a range of cAMP concentrations. (**D**) Reaction scheme used for modeling type II PKA regulation by CN. Each species within the scheme is numbered consistent with supporting data in figure in ***Supplementary file 1***. (**E–G**) Model simulations for protein mixtures activated with 1 μM cAMP are shown with the experimental data overlaid. Averaged values are shown for experimental data after pooling the data shown in ***Figure 3***. Responses are shown for type II PKA alone (**E**), with CN (**F**), and with both CN and AKAP79$_{c97}$ (**G**). A sample of the corresponding simulated responses are shown in grey. An 'error' threshold of 0.01 was used to accept curves as a good fit.

The online version of this article includes the following figure supplement(s) for figure 4:

**Source data 1.** Free C subunit calculations.

**Figure supplement 1.** Simulations of kinetic scheme species changes in concentration over time.

**Figure supplement 2.** Simulations of responses with different concentrations of cAMP.

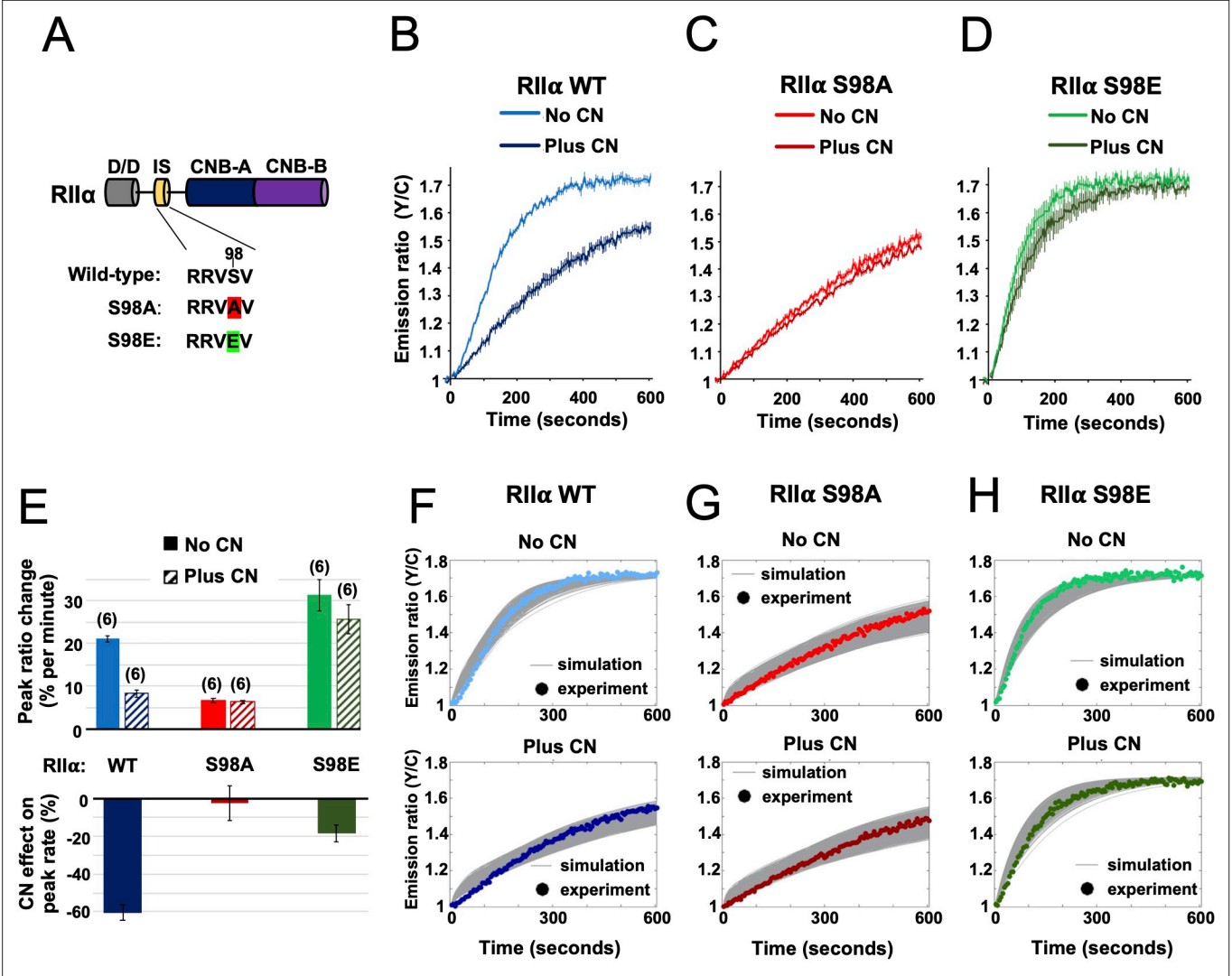

**Figure 5.** Characterization of RIIα IS phosphorylation site mutations. (**A**) RII subunit topology showing locations of the docking and dimerization domain (D/D, gray), inhibitor sequence (IS, yellow), and tandem cyclic nucleotide binding domains (dark and light blue). S98A (red) and S98E (green) mutations in the IS are highlighted. (**B–D**) Comparison of AKAR4 emission ratio changes following 5 mM ATP/1 µM cAMP activation of protein mixtures containing either WT (**B**), S98A (**C**), or S98E (**D**) RII. 1.03 µM RII $\beta$ was included in all cases. Measurements were collected either with or without CN in the reaction mixture. Averaged responses (± SE) are shown for WT RII with (dark blue) and without CN (light blue), S98A RII with (dark red) and without (light red) CN, and RII S98E with (dark green) and without (light green) CN. (**E**) The upper bar chart shows peak rates (calculated between 30 and 90 s) for the responses shown in panels b-d. The effect of including CN in the reaction mixture for each RII variant is shown in the lower bar chart. (**F–H**) Model predictions in the six conditions of panels b-d are shown in grey when simulating using the 'extended' model (see Materials and methods) and using the different parameter sets generated from the parameter estimation approach. The same parameters as retrieved using data shown in *Figure 4* were used as a starting point for the simulations, but parameter sets were filtered based on data collected with RII S98A. Model predictions are shown alongside the corresponding experimental data collected with either WT (**F**), S98A (**G**), or S98E (**H**) RII in the reaction mix.

The online version of this article includes the following figure supplement(s) for figure 5:

**Source data 1.** Rates of AKAR4 phosphorylation with mutant RIIα subunits.

**Figure supplement 1.** Space of parameters used in model fitting.

*Sanderson et al., 2016*), show that both AKAP150 anchoring sites are required for LTD induction. However, such approaches cannot distinguish between CN targeting to pRII subunits versus other substrates. If direct suppression of PKA activity by CN is essential for LTD induction, we reasoned that mutation of the IS phospho-acceptor S98 (*Figure 5A*) in the predominant RIIα isoform would be expected to disrupt LTD induction in CA1 neurons. To confirm this presupposition before undertaking experiments in neurons, we re-ran AKAR4 experiments at 1 µM cAMP substituting in either

S98A or S98E RIIα. For each RIIα variant (*Figure 1—figure supplement 1H*), we compared responses with or without CN, with WT RIIβ and AKAP79$_{c97}$ present in all cases. For WT RIIα, addition of CN to the mixture decreased the peak rate of AKAR4 phosphorylation from 21.02 ± 0.76 (light blue, *Figure 5B*) to 8.24% ± 0.79 % per minute (dark blue). Substituting in RIIα S98A generated slow rates of AKAR4 phosphorylation in both cases (6.30% ± 0.44 % per min with CN, and 6.67% ± 0.56 % without, *Figure 5C*). Conversely, the peak rate of AKAR4 phosphorylation was high regardless of the presence of CN for the S98E RIIα variant (31.30% ± 3.60 % per min without CN; 25.65% ± 3.44 % with CN, *Figure 5D*). Together, this data indicates that substituting in either mutant of RIIα in neurons would be expected to reduce LTD induction in neurons if direct suppression of PKA by CN is required in LTD induction (*Figure 5E*).

Before moving on to experiments in neurons, we used the data collected with RIIα variants to test the accuracy of our kinetic modeling. We ran simulations assuming that the S98A and S98E variants of RIIα would behave like dephosphorylated and phosphorylated forms of the regulatory subunit. Broadly, the simulations were in line with our experimental data and predicted that addition of CN would reduce PKA activity substantially more in the WT but not RIIα mutant conditions (*Figure 4—figure supplement 1*), with low and high PKA activities regardless of CN concentration for the S98A and S98E variants, respectively. The model predictions for the extent by which AKAR4 phosphorylation was depressed in the RIIα S98A system were, however, spread out depending on the specific parameter set (column 4–5, *Figure 4—figure supplement 1*). This implies that the WT data we used to constrain the model were not sufficient to precisely constrain the dynamics specifically for the unphosphorylated RII sub-system (right square, *Figure 4D*) To understand the characteristics of those parameter sets that also reproduced the RIIα S98A behavior, we filtered the parameter sets returned by the parameter estimation approach into two classes depending on whether they fit closely (blue, *Figure 4—figure supplement 1*) or not (red) to the acquired mutation data, yielding 526 parameter sets that fit closely to both the WT and mutation data. A pairwise coordinate plot (see *Figure 5—figure supplement 1A*) shows that, except for a few parameters, the two classes do not appear to be visually distinct with regard to kinetic rates. However, analysis and subdivision of the eight model dissociation constants (K$_D$'s) reveals interesting relationships (*Figure 5—figure supplement 1B*). Notably, as shown by the histograms and scatterplot for the K$_D$ for interaction between RII-C and cAMP (K$_D$56), and RII-cAMP and C (K$_D$76), K$_D$56 is most often relatively low within its range paired with a relatively high K$_D$76 (*Figure 5—figure supplement 1C & D*) when accurately mimicking the biological workings of the PKA sub-system. This behavior may ensure that sufficient C subunit is released with increasing cAMP in our model when the kinetics are restrained to the unphosphorylated RII sub-system, i.e when the RIIα S98A mutation is introduced. Overall, simulations using unfiltered (top row, *Figure 4—figure supplement 1*) and filtered (*Figure 5F–H*) parameter sets show that the kinetic model closely reproduces the experimental data, especially when further constrained using data collected with RIIα S98A. Furthermore, the constrained simulations reproduce the experimental data collected at different cAMP concentrations (*Figure 4—figure supplement 2*). Taken together, experiments and simulations with S98A and S98E variants of RIIα show that either of these mutations should prevent AKAP79 and CN from switching C subunit capture from the left-hand square subsystem to the faster right-hand square (*Figure 4D*). Therefore, either substitution would be expected to reduce LTD induction if the mechanism is important in vivo.

## Disruption of RIIα phosphorylation in CA1 neurons impedes chemical LTD

To enable neuronal RIIα replacement experiments, we generated lentiviruses for shRNA-mediated knockdown of endogenous RIIα and simultaneous expression of shRNA-resistant RIIα variants in tandem with GFP. The lentiviruses contain an H1 promoter for expression of a highly effective shRNA targeted to RIIα (*Figure 6A*). A UbC promoter drives expression of replacement RIIα sequences in tandem with GFP, with an internal ribosome entry sequence (IRES2) between the coding sequences of the two proteins enabling expression of GFP. We validated the lentiviruses in dissociated rat primary hippocampal neurons by comparing the efficacy of five different lentiviruses. On day 7 in vitro (DIV7), we infected with control lentiviruses expressing either scrambled or shRIIα RNA, or with complete viruses for replacement of endogenous RIIα with either WT, S97A, or S97E (RIIα in rat is equivalent to S98 in human RIIα). Neuronal protein extracts were collected on DIV14, and analyzed using

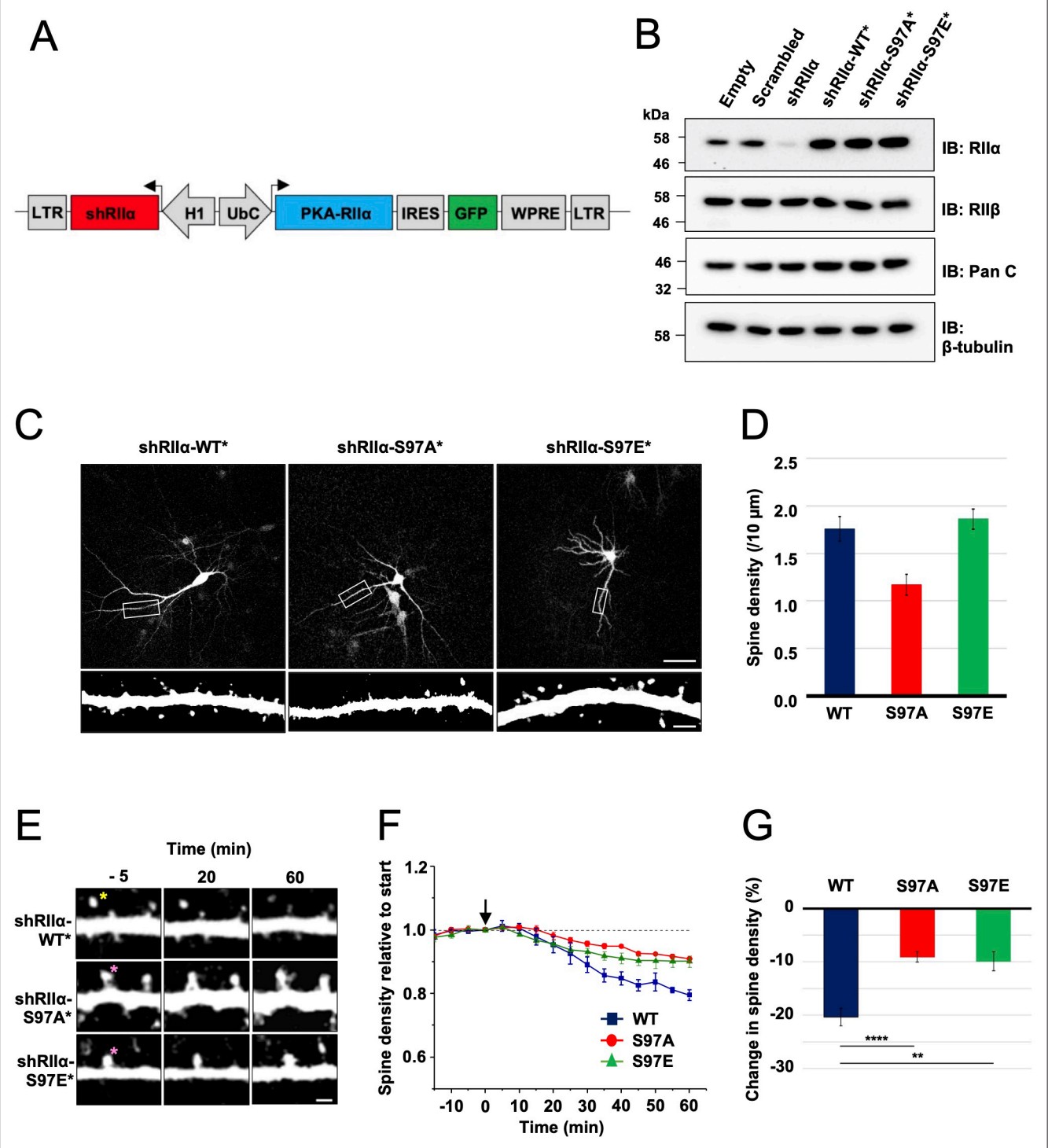

**Figure 6.** Lentivirus development and spine density imaging. (**A**) Schematic of the FUGW-H1-based lentiviral vector used to knock down and replace endogenous RIIα subunits in dissociated hippocampal cultures. (**B**) To validate lentiviruses, dissociated hippocampal neurons were infected on the seventh day in vitro (DIV7). Immunoblots are shown comparing neuronal extracts collected on DIV14 after infection with no virus, virus expressing scrambled shRNA only, shRIIα only, and the three complete lentiviruses for knockdown/replacement with either WT, S97A, or S97E RIIα. (**C**) Representative live-cell images of lentivirus-infected primary hippocampal neurons at DIV14 expressing either WT, S97A, or S97E RIIα. Scale bars correspond to 50 μm (upper panels) and 5 μm (lower panels). (**D**) Average spine density on hippocampal dendrites following lentiviral replacement of endogenous RIIα. Data were averaged from 106 (WT), 97 (S97A), and 113 (S97E) neurons derived from seven rats for each condition, and are represented as mean ± SE. Conditions were compared using one-way ANOVA with Turkey post-hoc tests. (**E**) Representative live-cell images showing dendritic spines in primary hippocampal neurons expressing either WT, S97A, or S97E replacement RIIα at three points before and after chem-LTD

*Figure 6 continued on next page*

*Figure 6 continued*

(scale bar = 2.5 µm). Chem-LTD was induced at t = 0 with 20 µM NMDA for 3 min. The yellow asterisk indicates a spine that disappeared over the course of the protocol whereas the pink asterisks indicate spines that did not. (**F**) Plot showing average changes in spine density (± SE) in primary hippocampal neurons expressing either WT (dark blue), S97A (red), or S97E (green) RIIα. (**G**) Average changes in spine density± SE 1 hr after induction of chem-LTD are shown for neurons expressing WT (dark blue, n = 5), S97A (red, n = 5), and S97E (green, n = 4) RIIα variants as shown in the preceding two panels. Statistical comparisons were performed by two-way ANOVA followed by Bonferroni's post-hoc test. **p < 0.01, ***p < 0.001.

The online version of this article includes the following figure supplement(s) for figure 6:

**Source data 1.** Spine density quantitation.

immunoblotting. Anti-RIIα immunoblotting (top row, *Figure 6B*) confirmed effective suppression of endogenous RIIα with shRIIα (lane 3) but not scrambled RNA (lane 2), and strong expression of the replacement sequences (lanes 4–6). Expression of PKA C (row 2, *Figure 6B*) and RIIβ subunits (row 3) was not affected by lentiviral infection in any case. Blocking PKA activity with H89 is known to prevent growth of new spines, whereas stimulating PKA with forskolin increases spine formation (*Kwon and Sabatini, 2011*). Replacing RIIα with the S97A variant – which has lower PKA activity regardless of CN activity (*Figure 5F*) – would therefore be expected to lead to a reduction in spines. To test this, we imaged dendritic spines on primary hippocampal neurons expressing either WT (left panel, *Figure 6C*), S97A (middle panel), or S97E (right panel) RIIα. Consistent with a role for PKA in spino-genesis, spine density was reduced by 33.5 % (p = 0.002) in neurons expressing the S97A variant to 1.17 ± 0.11 spines per 10 µm compared to 1.76 ± 0.12 for WT RIIα. Spine density for the S97E variant was similar to WT at 1.86 ± 0.11 spines/10 µm (*Figure 6D*).

To test whether the two substitutions at RIIα S97 affect LTD, we monitored changes in dendritic spine number during chemical LTD – a model of long-term synaptic depression that can be applied in dissociated neuronal cultures. Bath application of 20 µM NMDA for 3 min triggered a steady reduction in spine density (*Figure 6E*, top row) in neurons expressing WT RIIα as expected (*Zhou et al., 2004*), reaching a 20.4% ± 1.6%% reduction in spines after one hour (blue, *Figure 6F*). In comparison, spine loss was attenuated in neurons expressing either the S97A (*Figure 6E*, middle row) or S97E (bottom row) RIIα variants. Spine numbers were reduced by only 9.07% ± 0.96 % in neurons expressing RIIα S97A (red line, *Figure 6F*), and by 9.90% ± 1.8 % for the S97E variant (green line). The residual LTD in both conditions may correspond to action of CN on substrates other than pRII subunits, and limited suppression of PKA activity through CN dephosphorylation of the relatively small number of WT RIIβ subunits that are present in all cases. Overall, attenuation of spine loss in neurons expressing either S97A (p = 0.00046) or S97E ( = 0.0014) RIIα compared to WT subunits is consistent with an important role for direct PKA activity suppression by CN during the induction of LTD.

## Discussion

The observations in this study support a revised mechanism for CN-mediated long-term depression in CA1 model synapses. AKAP79/150 is critical for anchoring PKA in dendritic spines (*Tunquist et al., 2008*; *Weisenhaus et al., 2010*) through association with RII subunits, which are the predominant neuronal PKA subunit in ~11 fold molar excess of C subunits in the CA1 neuropil (*Figure 2D*). Imaging studies (*Ilouz et al., 2017*; *Weisenhaus et al., 2010*) are consistent with our quantitative immunoblotting data, which show that RIIα is the major RII isoform in the CA1 neuropil. pRII dephosphorylation is limited prior to Ca$^{2+}$ stimulation (*Figure 7A*), enabling a tonic level of dissociated C subunits sufficient to basally phosphorylate postsynaptic substrates in dendritic spines such as GluA1 subunits of AMPA-type glutamate receptors (*Bear, 2003*). LTD is brought about by CN (*Mulkey et al., 1994*), which is activated by Ca$^{2+}$ entering spines through NMDA-type glutamate receptors (*Figure 7B*). AKAP79/150 contains a 'PIAIIIT' CN anchoring motif that is necessary for LTD (*Jurado et al., 2010*; *Sanderson et al., 2012*). In addition to potentially targeting CN to postsynaptic substrates including GluA1 subunits, the PIAIIIT anchoring motif positions CN adjacent to pRII subunits where it can efficiently dephosphorylate them (*Figure 7B*). This enables CN to increase the concentration of dephosphory-lated RII species (blue spheres in the kinetic scheme shown in *Figure 7B*) thereby directly suppressing PKA activity by increasing the rate of PKA C subunit capture. Consistent with this mechanism, blocking regulation of RII phosphorylation state by introducing mutations that mimic either the phosphorylated or dephosphorylated forms of the IS reduces LTD in cultured hippocampal neurons.

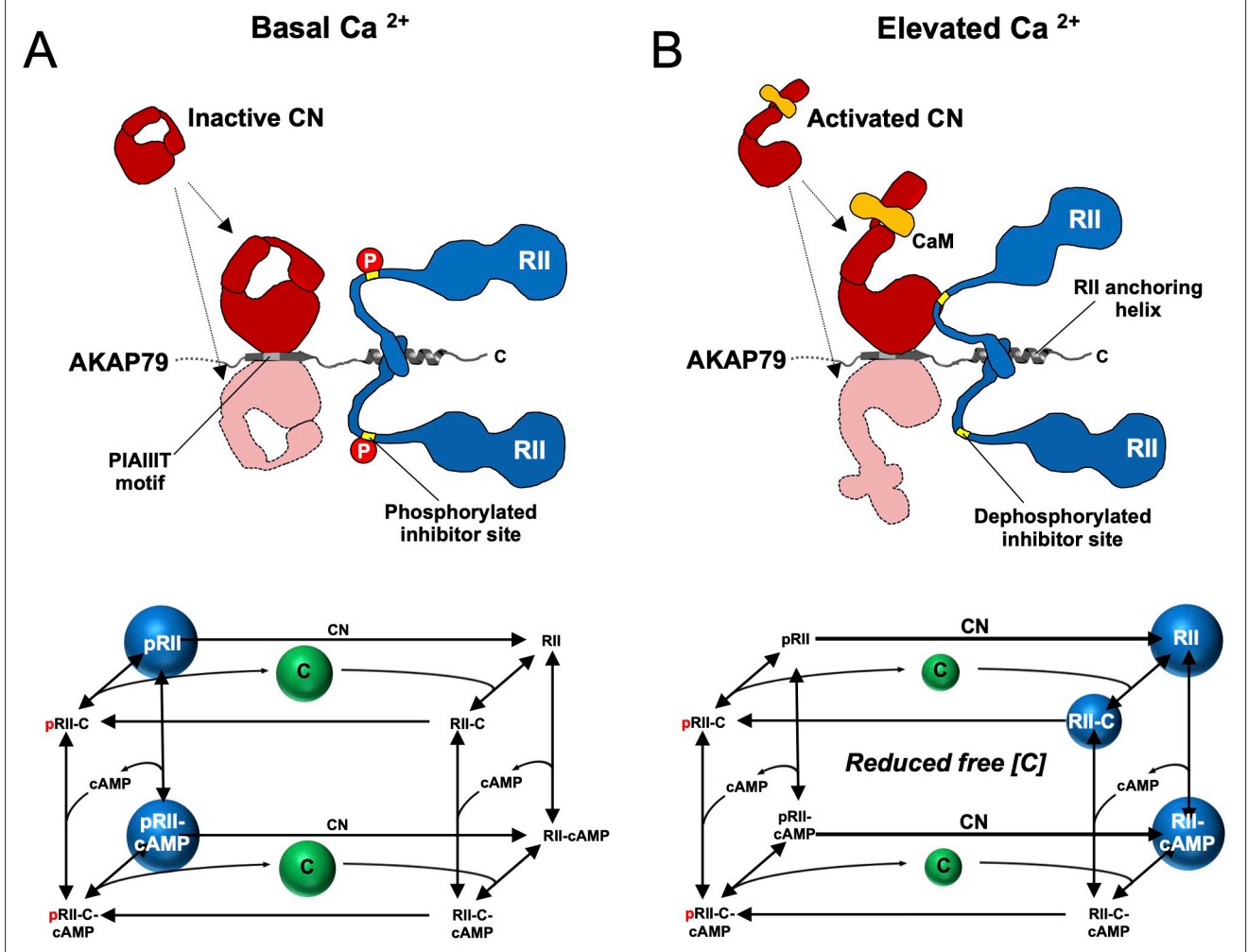

**Figure 7.** Summary model of PKA suppression by CN within the AKAP79 complex. Structural and kinetic models (upper and lower panels, respectively) of signaling within the AKAP79 complex are shown under conditions of either low (**A**) or elevated $Ca^{2+}$ (**B**). Elevated $Ca^{2+}$ triggers CN (red) dephosphorylation of pRII (blue) which shifts C subunit capture from the left-hand square of the kinetic scheme to the right-hand square which features dephosphorylated forms of RII. The overall effect is a reduction in the concentration of free C subunits. The most abundant forms of RII under the two conditions are highlighted by blue spheres.

Our discovery that CN can directly suppress PKA activity in the AKAP79 complex reconciles three aspects of AKAP79 structure and function that had been enigmatic and paradoxical. First, previous studies showed that AKAP79 acts as a weak inhibitor of CN towards peptide substrates including a 20-mer peptide encompassing the phosphorylated RII IS (*Coghlan et al., 1995*; *Kashishian et al., 1998*), apparently at odds with the functional requirement for the anchoring protein in targeting CN to bring about LTD. We show that the key substrate for CN is likely to be full-length pRII subunits, and that in fact AKAP79 enhances the activity toward pRII at physiological concentrations by more than 10-fold. A second enigmatic feature of AKAP79 is its CN anchoring motif, PIAIIIT, which includes an additional central residue compared to the typical PxIxIT motif (*Roy and Cyert, 2009*). In a crystal structure of CN in complex with a peptide corresponding to AKAP79 positions 336–346, the additional leucine supports simultaneous binding of two copies of CN on either side of the motif (*Li et al., 2012*). Native mass spectrometry measurements of a purified AKAP79-CN-CaM-RIIα D/D complex also support a stoichiometry of 2 CN to 1 AKAP79 (*Gold et al., 2011*), although solution measurements indicate that when full-length RII subunits are bound to AKAP79, only one copy of CN can bind at a time (*Li et al., 2012*; *Nygren et al., 2017*). One possible explanation for this behavior is that CN binds transiently to either side of the AKAP79 PIAIIIT motif enabling it to access both protomers of RII anchored to AKAP79 for efficient pRII dephosphorylation (cartoon representations in *Figure 7*).

This idea is consistent with data showing that mutating the PIAIIIT motif to a high-affinity canonical PxIxIT motif impairs the function of the phosphatase (*Li et al., 2012*), although it should be noted that it is not possible to determine whether two-sided CN binding to AKAP79 is necessary using the data presented here. Third, existing models of AKAP79 function assume that CN anchored to AKAP79 overcomes PKA phosphorylation at substrates with no reduction in PKA phosphorylation rate. In our revised mechanism, CN directly suppresses PKA activity when removing phosphate from substrates primed by PKA thereby avoiding energetically costly ongoing futile cycling of phosphorylation and dephosphorylation by PKA and CN at these sites.

A challenge in the future will be to understand how the mechanism uncovered here relates to the full complexity of AKAP79 function. AKAP79 is directly regulated by $Ca^{2+}$/CaM, which binds to a 1-4-7-8 hydrophobic motif (*Patel et al., 2017*) starting at position W79. Binding of $Ca^{2+}$/CaM releases AKAP79 from the postsynaptic membrane (*Dell'Acqua et al., 1998*) and alters the conformation of the signaling complex by triggering formation of a second interface between CN and AKAP79 that involves an LxVP-type motif in AKAP79 (*Gold et al., 2011*; *Nygren et al., 2017*). Furthermore, metal ions including $Ca^{2+}$ alter rates of substrate binding and product release from PKA C subunits (*Knape et al., 2015*; *Zhang et al., 2015*). Therefore, it will be important to understand the sensitivity of CN suppression of PKA activity to $Ca^{2+}$ signals. Membrane targeting sequences regulate several components of the AKAP79 signaling complex. Myristylation of C subunits is important for limiting their diffusion rate in dendritic spines and concentrating PKA activity at the cell membrane (*Tillo et al., 2017*; *Xiong et al., 2021*). Localization of AKAP79 is also regulated by palmitoylation at C36 and C139 (*Delint-Ramirez et al., 2011*; *Keith et al., 2012*). Palmitoylation is required for endosomal localization of AKAP79, and AKAP79 depalmitoylation and synaptic removal is additionally regulated by CaMKII (*Woolfrey et al., 2018*). Our work suggests that removal of AKAP79 from synapses might be synchronized with accumulation of inhibited C subunits in the AKAP79 complex. Given that RII subunits greatly outnumber C subunits, movement of C subunits between different RII sub-populations, including RII subunits anchored to MAP2 in dendritic shafts (*Tunquist et al., 2008*), should also be considered along with PDEs that can terminate cAMP signals with high spatiotemporal precision (*Bock et al., 2020*; *Tulsian et al., 2017*). Non-dissociative activation of anchored type RII PKA has been put forward as an alternative mechanism to explain localised PKA activity (*Smith et al., 2017*). Current evidence indicates that C subunits do dissociate in neurons upon elevation of cAMP (*Gold, 2019*; *Mo et al., 2017*; *Tillo et al., 2017*), but it is important to note that suppression of PKA by pRII dephosphorylation is compatible with non-dissociative models of PKA activation and this might occur in certain physiological settings. AKAP79 is a highly multivalent protein – other notable documented interaction partners include protein kinase C (*Hoshi et al., 2010*) and the $Ca^{2+}$-activated cyclase AC8 (*Baldwin and Dessauer, 2018*; *Zhang et al., 2019*). Oscillations of $Ca^{2+}$, cAMP, and PKA activity have been observed in pancreatic β-cells (*Hinke et al., 2012*; *Ni et al., 2011*), and knockout of AKAP150 leads to the loss of cAMP oscillations in β-cells upon stimulation with insulin (*Hinke et al., 2012*). CN dephosphorylation of pRII subunits bound to AKAP79 is likely to play a role in oscillatory patterns of PKA activity, and it will be important to understand how this mechanism underlies responses to short-lived and oscillatory changes in $Ca^{2+}$ and cAMP concentration.

In this combined experimental-computational study, we focused on AKAP79 signaling in dendritic spines on the basis that this could serve as a prototype for understanding a potentially widespread non-canonical mechanism for altering PKA. In addition to its role in dendritic spines, AKAP79 regulates many different membrane channels and receptors following $Ca^{2+}$ influx through a variety of sources, and the mechanism that we have uncovered here is likely to at least extend to these additional contexts. For example, AKAP79 underlies $GABA_A$ receptor regulation during LTD of GABAergic synapses (*Dacher et al., 2013*), and it positions PKA and CN for regulation of TRPV channels (*Zhang et al., 2008*), Kv7 channels (*Zhang and Shapiro, 2012*), and β-adrenergic receptors (*Houslay and Baillie, 2005*). AKAP79 is also necessary for NFAT dephosphorylation following $Ca^{2+}$ entry through both L-type calcium channels (*Wild et al., 2019*) and the store-operated $Ca^{2+}$ channel ORAI1 (*Kar et al., 2014*). The RII IS phosphorylation site is conserved throughout the animal kingdom, and co-anchoring of phosphatases alongside PKA is a feature of several AKAP complexes (*Redden and Dodge-Kafka, 2011*). Future investigations may therefore explore whether additional anchoring proteins are able to direct CN – or other cellular phosphatases – for direct suppression of PKA activity.

# Materials and methods

**Key resources table**

| Reagent type (species) or resource | Designation | Source or reference | Identifiers | Additional information |
|---|---|---|---|---|
| Strain, strain background (*Escherichia coli*) | TOP10 chemically competent | Life Technologies | Cat# C404003 | |
| Strain, strain background (*Escherichia coli*) | BL21 (DE3) | Thermo Fisher Scientific | Cat# EC0114 | |
| Strain, strain background (*Escherichia coli*) | BL21 Tuner (DE3) pLysS | Merck | Cat# 70,624 | |
| Strain, strain background (*Escherichia coli*) | BL21 Star (DE3) | Thermo Fisher Scientific | Cat# C601003 | |
| Strain, strain background (*Escherichia coli*) | *Stbl3* | Thermo Fisher Scientific | Cat# C737303 | |
| Cell line (*Homo-sapiens*) | HEK293 | Horizon Discovery LTD | Cat# HCL3417 | Myocplasma tested. |
| Cell line (*Homo-sapiens*) | HEK293T | ATCC | Cat# CRL-3216 | Myocplasma tested. |
| Biological sample (*Rattus norvegicus*) | Sprague Dawley | UCL breeding colony | Not applicable | |
| Antibody | (Mouse monoclonal) anti-PKA RII $\alpha$ | BD Biosciences | Cat# 612243; RRID:AB_399566 | (0.8 µg/mL) |
| Antibody | (Mouse monoclonal) anti-PKA RIIβ | BD Biosciences | Cat# 610626; RRID:AB_397958 | (0.8 µg/mL) |
| Antibody | (Mouse monoclonal) anti-PKA C (pan) | BD Biosciences | Cat# 610981; RRID:AB_398294 | (0.5 µg/mL) |
| Antibody | (Mouse monoclonal) anti-PKA RI (pan) | BD Biosciences | Cat# 610166; RRID:AB_397567 | (0.8 µg/mL) |
| Antibody | (Rabbit monoclonal) anti-PKA phospho-RII $\alpha$ | Abcam | Cat# ab32390; RRID:AB_779040 | (0.8 µg/mL) |
| Antibody | (Rabbit polyclonal) anti-GFP | Sigma Aldrich | Cat# SAB4301138; RRID:AB_2750576 | (0.5 µg/mL) |
| Antibody | (Mouse monoclonal) anti-β-tubulin | Biolegend | Cat# 903401; RRID: AB_2565030 | (0.5 µg/mL) |
| Antibody | Goat anti-rabbit HRP-linked secondary antibody | Cell Signalling Technology | Cat # 7,074 S; RRID:AB_2099233 | (1 µg/mL) |
| Antibody | Goat anti-mouse IgG (H + L) poly-HRP secondary antibody | Thermo Fisher Scientific | Cat# 32230; RRID:AB_1965958 | (1 µg/mL) |
| Recombinant DNA reagent | pIRES2-EGFP | Clontech | Cat# 6029–1 | |
| Recombinant DNA reagent | pFUGW-H1 | Sally Temple lab/ Addgene | Cat# 25870; RRID:Addgene_25870 | Lentiviral entry vector. |
| Recombinant DNA reagent | pFUGW-shRII $\alpha$ -RII $\alpha$ *-WT-IRES-EGFP | This study | Not applicable | Lentivral entry vector. Dr. Matthew G. Gold (University College London) |
| Recombinant DNA reagent | pFUGW-shRII $\alpha$ -RII $\alpha$ *-S97A-IRES-EGFP | This study | Not applicable | Lentiviral entry vector. Dr. Matthew G. Gold (University College London) |

*Continued on next page*

*Continued*

| Reagent type (species) or resource | Designation | Source or reference | Identifiers | Additional information |
|---|---|---|---|---|
| Recombinant DNA reagent | pFUGW-shRII $\alpha$ -RII $\alpha$ *-S97E-IRES-EGFP | This study | Not applicable | Lentivral entry vector. Dr. Matthew G. Gold (University College London) |
| Recombinant DNA reagent | pCMVdR8.74 & pMD2.G plasmids | Didier Trono lab/ Addgene | Cat# 12259; RRID:Addgene_12259 | Lentiviral packaging vectors |
| Recombinant DNA reagent | pcDNA3.1-AKAR4-NES | Jin Zhang lab/Addgene | Cat# 64727; RRID:Addgene_64727 | |
| Chemical compound, drug | Lipofectamine 2000 | Thermo Fisher Scientific | Cat# 11668019 | |
| Chemical compound, drug | DMEM, high glucose, pyruvate | Thermo Fisher Scientific | Cat # 41966029 | |
| Chemical compound, drug | Trypsin | Thermo Fisher Scientific | Cat# 25300054 | |
| Chemical compound, drug | Penicillin/ Streptomycin | Thermo Fisher Scientific | Cat# 15140122 | |
| Chemical compound, drug | GlutaMAX | Thermo Fisher Scientific | Cat# 35050061 | |
| Chemical compound, drug | DPBS, no calcium, no magnesium | Thermo Fisher Scientific | Cat# 14190144 | |
| Chemical compound, drug | HBSS | Thermo Fisher Scientific | Cat# 14185045 | |
| Chemical compound, drug | Heat-inactivated horse serum | Gibco | Cat# 26050088 | |
| Chemical compound, drug | Neurobasal-A medium | Thermo Fisher Scientific | Cat# 10888022 | |
| chemical compound, drug | B27 supplement | Gibco | Cat# 17504044 | |
| chemical compound, drug | Poly-L-Lysine | Sigma Aldrich | Cat# P2636 | |
| chemical compound, drug | Boric acid | Sigma Aldrich | Cat# B6768-500g | |
| chemical compound, drug | Sodium tetraborate | Sigma Aldrich | Cat# 221,732 | |
| chemical compound, drug | cOmplete, Mini, EDTA-free Protease Inhibitor Cocktail | Roche | Cat# 11836170001 | |
| chemical compound, drug | PhosSTOP phosphatase inhibitor tablets | Roche | Cat# 4906845001 | |
| chemical compound, drug | Para-nitrophenylphosphate | Sigma Aldrich | Cat# N3254 | |
| software, algorithm | Origin | OriginLab | http://www.originlab.com/; RRID:SCR_014212 | |
| software, algorithm | Reader Control Software for FLUOStar Omega | BMG Labtech | https://www.bmglabtech.com/reader-control-software/ | |
| software, algorithm | MARS Data Analysis Software | BMG Labtech | https://www.bmglabtech.com/mars-data-analysis-software/ | |
| software, algorithm | Unicorn Start 1.1 Software for controlling AKTA start system | GE Healthcare | Cat# 29225049 | |
| software, algorithm | ImageJ (version 1.52) | NIH | RRID:SCR_003070 | |
| software, algorithm | NeuronStudio | *Rodriguez et al., 2008* | https://icahn.mssm.edu; RRID:SCR_013798 | |

## Protein Expression and Purification

Human PKA subunits were expressed and purified as described previously (*Walker-Gray et al., 2017*). GST-RIIα and GST-RIIβ were expressed in *Escherichia coli* BL21 Tuner (DE3) pLysS, and GST-Cβ in *E. coli* BL21 (DE3) grown in LB. In all cases, protein expression was induced by addition of 0.5 mM isopropyl β-D-1-thiogalactopyranoside (IPTG), and bacteria were harvested following overnight incubation at 20 °C. Cell pellets were thawed and sonicated in glutathione sepharose binding buffer (20 mM HEPES pH 7.5, 500 mM NaCl, 1 mM DTT, 0.5 mM EDTA, 1 mM benzamidine, 10 % glycerol) supplemented with 0.1 mg/mL lysozyme, and 0.1 % Igepal CA-630 (RII subunit preps only). Clarified lysates were incubated with glutathione sepharose 4B, and PKA subunits were eluted by overnight cleavage with PreScission protease thus removing N-terminal GST affinity tags. Finally, each subunit was purified using a HiLoad 16/600 Superdex 200 column connected in series with a GSTrap to remove residual GST using gel filtration buffer (20 mM HEPES pH 7.5, 150 mM NaCl, 5 % glycerol). S98A and S98E point mutations were introduced into RIIα subunits by site-directed mutagenesis (SDM) with primer pairs hS98A_F & R, and hS98E_F & R. RIIα variants were expressed and purified in the same way as the WT sequences.

Full-length human AKAP79 was cloned into pET28 using primers Nde1_AKAP79_1 and AKAP79_427_EcoRI for expression of N-terminally 6His-tagged protein. AKAP79 was expressed in 4 L BL21 Star (DE3) cells by overnight incubation at 37 °C in auto-induction media (AIM). PBS-washed bacterial pellets were resuspended in Talon binding buffer (30 mM Tris pH 8.0, 500 mM NaCl, 10 mM imidazole, 1 mM benzamidine) supplemented with 0.1 mg/mL lysozyme and one Complete EDTA-free protease inhibitor tablet (Roche) per 100 mL. Lysates were sonicated, clarified by centrifugation, and incubated with Talon Superflow resin for 2 hours prior to 3 × 10 mL washing in Talon binding buffer, and eluted with 2 × 2.5 mL Talon elution buffer (30 mM Tris, pH 7.0, 500 mM NaCl, 300 mM imidazole, 1 mM benzamidine). Eluted protein was exchanged into Q buffer A (20 mM Tris pH 8, 20 mM NaCl, 1 mM EDTA, 2 mM DTT) using a HiPrep 26/10 desalting column to enable purification using a 1 mL Resource Q column. Each variant was eluted using a NaCl/pH gradient with Q buffer A and a steadily increasing proportion of Q buffer B (20 mM Tris pH 7, 500 mM NaCl, 1 mM EDTA, 2 mM DTT). In the final step, peak fractions were pooled and buffer exchanged into gel filtration buffer. Residues 331–427 of AKAP79 were cloned into pET28 using primers Nde1_AKAP79_331 and AKAP79_427_EcoRI for expression of the fragment AKAP79$_{c97}$ bearing an N-terminal His tag. This construct was transformed into BL21 (DE3) cells, which were grown overnight at 37 °C in AIM. Lysis and metal affinity steps were as for full-length AKAP79 with the exception that Ni-NTA agarose (Life Technologies) was used in place of Talon resin. Following elution from Ni-NTA resin, the protein was purified by size exclusion using a HiLoad 16/600Superdex 200 pre-equilibrated in gel filtration buffer. To assemble complexes of full-length RII subunits and AKAP79$_{c97}$, mixtures of the purified proteins were incubated on ice in gel filtration buffer for 1 h with the AKAP fragment in a 2:1 molar excess. The complex was then separated from excess AKAP79$_{c97}$ by Superdex 200 size exclusion. pET28-AKAP79$_{c97}$ΔCN was generated by performing PCR with an earlier construct lacking residues 337–343 as the template (*Gold et al., 2011*), whereas the ΔPKA variant (lacking residues 391–400) was generated by SDM with primers ΔPKA_F & _R. The two AKAP79$_{c97}$ deletion mutants were expressed and purified in the same way as the WT protein.

Human CN was expressed from a bicistronic pGEX6P1 vector (*Gold et al., 2011*) in *E. coli* BL21 Tuner (DE3) pLysS cells. Protein expression was induced by overnight incubation at 37 °C in 4 L AIM. CN was purified following the same protocol as full-length PKA RII subunits, with the final size exclusion step performed using gel filtration buffer supplemented with 1 mM DTT. Human CaM was expressed and purified as described previously (*Patel et al., 2017*). Briefly, untagged CaM was expressed in *E. coli* BL21 (DE3) cells incubated overnight at 37 °C in AIM. CaM was initially purified using phenyl sepharose resin, then by ion exchange with a HiTrap Q HP column. Finally, CaM was exchanged into water and lyophilized prior to storage at –80 °C. Human PP1α (7-300) was expressed in BL21 (DE3) *E. coli* in LB media supplemented with 1 mM MnCl$_2$ and purified as described previously (*Kelker et al., 2009*). The PP1 expression vector was a gift from Wolfgang Peti (Addgene plasmid # 26566). This vector was co-transformed with pGRO7 plasmid encoding the GroEL/GroES chaperone (Takara). PP1 expression was induced with 0.1 mM IPTG after prior induction of chaperone expression using 2 g/L arabinose. Bacteria were incubated for 20 hours at 10 °C following IPTG induction before harvesting. PP1 was purified by affinity to Ni-NTA agarose (Qiagen) followed by size exclusion with a Superdex 200 column equilibrated in PP1 gel filtration buffer (25 mM HEPES pH 7.5, 500 mM NaCl, 1 mM MnCl$_2$, 10 % glycerol). For

AKAR4 purification, an 8His epitope tag was ligated into pcDNA3.1-AKAR4-NES vector (*Depry et al., 2011*) (Addgene cat no. 64727) at the C-terminus of the sensor immediately prior to the nuclear export site using primers EcoI_8HisNLS_XbaI and XbaI_8HisNLS_EcoRI. The vector was transfected into 20 × 10 cm dishes of HEK293T cells cultured in DMEM using lipofectamine-2000 (Thermo Fisher Scientific). Cells were collected after 3 days, washed in PBS, then lysed in Talon binding buffer supplemented with 0.5 % Igepal CA-630, and sonicated briefly. AKAR4 was purified by affinity to Ni-NTA agarose following the same procedure as for AKAP79, and eluted protein was exchanged into gel filtration buffer, and aliquoted before storage at –80 °C. All purification columns and resins were purchased from GE Health-care. All protein samples were concentrated using Vivaspin centrifugal concentrators (Sartorius). Denaturing gel electrophoresis was performed using NuPAGE 4%–12% Bis-Tris gels (Thermo Fisher Scientific), and protein concentrations were determined using the bicinchoninic acid (BCA) assay. Oligonucleotide primer sequences are listed in *Supplementary file 2*.

## Phosphatase Assays

For radioactive release assays, CN substrates were prepared by phosphorylating PKA RII subunits at the autoinhibitory site with PKA C subunit and ATP($\gamma$-$^{32}$P). To radiolabel RII$\alpha$, RII$\beta$, or the purified complexes of each isoform with AKAP79$_{c97}$, 50 µg of the relevant sample was incubated in 100 µL with phosphorylation buffer (20 mM HEPES pH 7.5, 150 mM NaCl, 100 µM cAMP, 5 mM MgCl$_2$, 0.03 µg/µL C subunit) supplemented with 42 pmol [$^{32}$P-$\gamma$]-ATP at 3,000 Ci/mmol and 10 µM cold ATP. After 15 min incubation at 30 °C, reactions were supplemented with 10 µM additional cold ATP. Following 15 min further incubation, reactions were finally supplemented up to 1 mM cold ATP for 10 min further incubation. $^{32}$P-labelled protein was immediately separated from free $^{32}$P using Sephadex G-25 Medium equilibrated in phospho-substrate storage buffer (20 mM HEPES pH 7.5, 150 mM NaCl, 10 % glycerol, 0.1 mM EDTA). Additional cold phospho-labelled substrates were prepared using scaled-up reactions with 1 mM cold ATP for 30 min at 30 °C.

Phosphatase assays using $^{32}$P-labelled substrate (final volume 50 µL per assay) were prepared by first mixing appropriate dilutions of pRII substrates and CN on ice in dilution buffer (25 mM Na HEPES pH 7.5, 150 mM NaCl) to a final volume of 35 µL. 10 µL of reaction buffer (25 mM Na HEPES pH 7.5, 150 mM NaCl, 25 mM MgCl$_2$, 5 mM DTT, 0.5 mg/mL BSA, 1 mM EDTA) was then added before initiation of CN activity by addition of 5 µL activator mix (25 mM Na HEPES pH 7.5, 150 mM NaCl, 10 mM CaCl$_2$, 50 µM CaM). Assays was terminated after 30–60 s at 30 °C by addition of 350 µL 30 % trichloroacetic acid (TCA). Samples were then incubated on ice for 1 h, and protein was pelleted by centrifugation at 21,360× *g* for 15 min at 2 °C. The separated supernatant and pellet were analyzed using a Beckman LS 6000SC scintillation counter to determine the fraction of phosphate released from the pRII substrate. Reaction conditions were optimized so that less than 10 % pRII was dephosphorylated in each assay. Assays were generally performed with 10 nM CN and terminated after 30 s, with the exception of measurements for pRII$\alpha$ and pRII$\beta$ (black lines, *Figure 1E & F*) where 60 s reactions containing 100 nM CN were used.

For pNPP hydrolysis assays, para-nitrophenol (pNP) production was monitored continuously by measuring absorbance at 405 nm in a FLUOstar Omega microplate reader. Each 50 µL assay contained 5 µL of 10 x pNPP reaction buffer (100 mM Tris, pH 8.0, 100 mM NaCl, 10 mM CaCl2, 1 mg/ml of BSA, 60 mM MgCl2, 10 mM DTT), and 35 µL solution containing proteins at the appropriate concentrations in pNPP dilution buffer (100 mM Tris pH 8.0, 100 mM NaCl). Assays were performed with 200 nM CN, and 5 µM CaM where appropriate. Reactions were initiated by addition of 10 µL pNPP (Merck) to a final concentration of 5 mM, and pNP production was monitored at 22 °C for 1 hour at 1 minute intervals. For assays using phosphopeptide substrate, 19-mer pRII was synthesised by Biomatik at >95 % purity. Each 50 µL assay contained 5 µL of 10 x phosphopeptide reaction buffer (25 mM Na HEPES pH 7.5, 150 mM NaCl, 25 mM MgCl2, 5 mM DTT, 0.5 mg/mL BSA, 1 mM EDTA), and 30 µL solution containing proteins at the appropriate concentrations in phosphopeptide dilution buffer (25 mM Na HEPES pH 7.5, 150 mM NaCl). Assays were performed with 100 nM CN, and 3 µM CaM where appropriate. Assays were initiated by addition of pRII phosphopeptide to a final concentration of 40 µM, and terminated by addition of 50 µL Biomol Green solution (Enzo Life Sciences) following 3 min incubation at 22 °C. Free phosphate concentration was determined by measuring absorbance at 624 nm in the FLUOstar Omega microplate reader.

## Quantitative Immunoblotting of CA1 Neuropil Extracts

Hippocampal slices were prepared from 18 day old male Sprague-Dawley rats. Rats were euthanized by cervical dislocation and 350 µm-thick hippocampal slices were collected using a Leica VT1200S microtome in ice-cold sucrose-based saline (189 mM sucrose, 10 mM glucose, 3 mM KCl, 5 mM MgSO$_4$, 26 mM NaHCO$_3$, 1.25 mM NaH$_2$PO$_4$, 0.1 mM CaCl$_2$, pH 7.4) saturated with 95 % O$_2$/5 % CO$_2$. Slices were next transferred to a storage chamber filled with artificial cerebrospinal fluid (aCSF; 124 mM NaCl, 3 mM KCl, 24 mM NaHCO$_3$, 1.25 mM NaH$_2$PO$_4$, 1 mM MgSO$_4$, 10 mM glucose, 2 mM CaCl$_2$, pH 7.4) saturated with 95 % O$_2$/5 % CO$_2$ first for one hour at ~31 °C and at room temperature thereafter. For micro-dissection, slices were transferred onto a pre-chilled Sylgard-coated 90 mm petri dish atop a dry ice/ethanol bath. The CA1 neuropil layer was micro-dissected using an angled micro-knife (*Cajigas et al., 2012*) by first cutting along the borders of the stratum pyramidale/stratum radiatum and the stratum lacunosum moleculare/hippocampal fissure. Subsequent lateral cuts at the CA2-CA1 and subiculum-CA1 borders completed the rectangular micro-slices. Micro-dissected neuropil slices were immediately snap frozen in liquid nitrogen and stored at –80 °C. To extract protein, neuropil slices (~ 15 per animal) were first pulverized with a micro-pestle then resuspended in a final volume of 300 µL extraction buffer (50 mM Tris-HCl, 50 mM NaF, 10 mM EGTA, 10 mM EDTA, 0.08 mM sodium molybdate, 5 mM sodium pyrophosphate, 1 mM penylmethylsulfonyl fluoride, 0.5 % mM Igepal CA-630, 0.25% mM sodium deoxycholate, 4 mM para-nitrophenylphosphate, cOmplete EDTA-free protease inhibitors and PhosStop phosphatase inhibitors (Roche) at one tablet each per 50 mL). The homogenate was sonicated briefly (30 s at 20 MHz) then clarified by centrifugation at 21,130 x *g* (15 min at 4 °C). Total protein concentration in each extract was determined by BCA assay. Quantitative immunoblotting was performed as described previously (*Walker-Gray et al., 2017*) using anti-PKA subunit primary antibodies purchased from BD Biosciences. HRP-conjugated secondary antibodies were detected with WesternBright ECL chemiluminescent HRP substrate using a ImageQuant imaging unit (GE Healthcare). Band intensities for reference protein standards and neuropil extracts were calculated in ImageJ. For each immunoblot, a reference curve was generated by fitting reference protein concentrations and band intensities to a Hill function (with typical $R^2$ coefficients > 0.99) using iterative least squares refinement with the Levenberg-Marquardt algorithm in Origin (OriginLab). PKA subunit concentrations in neuropil extracts were determined by cross-referencing to reference curves derived from the same immunoblot.

## AKAR4 Measurements

AKAR4 fluorescence measurements were performed using black-walled 96-well plates in a FLUOstar Omega microplate reader (BMG Labtech) equipped with a 430 nm excitation filter, and 485 nm/520 nm emission filters. Each 50 µL reaction contained 35 µL proteins mixed in dilution buffer (20 mM HEPES pH 7.5 and 100 mM NaCl) including AKAR4 reporter (0.2 µM final concentration in all cases) and 5 µL of 10 x reaction buffer (20 mM Na HEPES pH 7.5, 100 mM NaCl, 10 mM DTT, 100 mM MgCl$_2$, 10 mM CaCl$_2$, 0.5 % Igepal CA-630). After three baseline measurements, PKA phosphorylation was initiated by addition of 10 µL solution containing ATP and the desired concentration of cAMP using two injectors built into the plate reader. One injector was primed with ATP solution (20 mM Na HEPES pH 7.5, 100 mM NaCl, 25 mM ATP) and the other with ATP/cAMP solution (20 mM Na HEPES pH 7.5, 100 mM NaCl, 25 mM ATP, 2.5 or 10 µM cAMP) so that different proportions of the two injectors could be used to vary the final cAMP concentration. Measurements were collected at 5 second intervals for a minimum of 10 minutes at 22 °C following injection of ATP. For every run, one control well was included in which AKAR4 was omitted from the protein mixture to enable baseline subtraction.

Phosphorylated AKAR4 (pAKAR4), for use in assays comparing PP1 and CN activity towards the reporter, was prepared by incubating 400 µg purified AKAR4 with 20 µg PKA C subunit in 1 mL AKAR4 phosphorylation buffer (25 mM Na Hepes pH 7.5, 150 mM NaCl, 10 mM MgCl$_2$, 5 mM ATP, 2 mM DTT). Following 30 min incubation at 30 °C, the phosphorylated reporter was exchanged into 25 mM Na Hepes pH 7.5 and 100 mM NaCl using Sephadex G-25 medium. In pAKAR4 dephosphorylation assays, each well contained 35 µL phosphatase at the appropriate concentration in dilution buffer mixed with 5 µL of 10 x reaction buffer. Reactions were initiated by injection of 10 µL AKAR4 solution to a final concentration of 0.2 µM, and measurements were collected at 5 s intervals for 15 minutes thereafter. For all AKAR4 assays, run parameters were set using Reader Control Software

for FLUOstar Omega, and measurements were analyzed using MARS Data Analysis Software (BMG Labtech). Aliquots of a single AKAR4 purification were used across all experiments.

## Kinetic Modeling

The model scheme of PKA activation is an updated and extended version of the one published by **Buxbaum and Dudai, 1989**. The model was simulated in a single reaction compartment devoid of any geometry as a system of chemical reactions mimicking the experimental conditions listed above. The individual chemical reactions were modeled as ordinary differential equation (ODE) using the chemical mass-action equation, as:

$$A+B \overset{k_f}{\underset{k_r}{\Longleftrightarrow}} AB$$

$$\frac{-d[A]}{dt} = \frac{-d[B]}{dt} = \frac{d[AB]}{dt} = k_f(x) = k_f[A][B] = k_r[AB]$$

In total, there were 16 chemical species and 16 reactions included in the model, incorporating mostly bi-molecular reactions with forward and backward reaction rates. Enzymatic reactions were represented by the three elementary steps of binding, dissociation and catalysis. All model variants were built using the MATLAB Simbiology toolbox (MathWorks). All reactions, along with initial concentrations of all chemical species and kinetic rates, are listed in **Supplementary file 1**.

PKA activation follows a sequential binding of four cAMP molecules to the PKA regulatory RII subunit holoenzyme followed by the release (or activation) of two active catalytic subunits (**Taylor et al., 2019**). However, the chosen modeling approach involved some simplifications: (1) The two RII subunits within the holoenzyme were assumed to behave independently – whereas in reality, some cooperativity is observed in PKA activation due to intra-dimeric contacts within the PKA holoenzymes (**Zhang et al., 2012**); (2) The two cAMP binding sites on the RII subunit were modelled as a single binding event such that binding of cAMP to RII/pRII is first order with respect to cAMP (**Hao et al., 2019**). This simplification was incorporated as our focus here was on understanding transitions between pRII and RII subunits and not the precise mechanism of cAMP activation; (3) The respective dephosphorylation parameters for both pRII and pRII bound to cAMP were assumed to be equal; (4) Rates of RII phosphorylation by bound C subunit were assumed to be equal irrespective of whether cAMP was bound to the regulatory subunit; (5) RIIα and RIIβ were assumed to behave similarly since isoform-specific differences were not the focus here. These simplifications were used to reduce the number of model parameters.

Parameters corresponding to the reactions involving dephosphorylation by CN were modified to represent the situations 'with' and 'without' AKAP79 (**Supplementary file 1**). In total eighteen different experimental AKAR4 responses were used to estimate the model parameters. Twelve corresponded to data shown in **Figure 3C & E** collected with either 0, 0.2, 1 or 2 μM cAMP activation: conditions with PKA (II)+ CaM either alone, with CN, or with both CN and AKAP79. The other six correspond to the calibration curves of C subunit interaction with AKAR4 (**Figure 3—figure supplement 1A**), which were used to estimate AKAR4 parameters that were kept frozen when the other model parameters were estimated. All parameters were estimated using an approximate Bayesian computation (ABC) approach, which included copulas for merging of different experimental data sets (**Eriksson et al., 2019**). A Bayesian approach was used over optimization for a single parameter set, to account for the uncertainty in the parameter estimates, and that more than one set of parameters could fit the data. The result is thus described using distributions for possible parameter values, rather than single values. Initial prior knowledge about the possible parameter values using data from this study, and previously published work from other groups (**Buxbaum and Dudai, 1989**; **Isensee et al., 2018**; **Moore et al., 2003**), was used to initiate the parameter fitting (details in **Supplementary file 1**). To account for parameter uncertainty, a log uniform prior distribution for the ABC-method was used. Many of the parameters were set to have a 'prior' range which varied three orders of magnitude from a default parameter value (black bar in **Figure 5—figure supplement 1A**), which ensured that different parameter values adopted in previous studies were included in the prior range. Simulations were started with initial conditions mimicking the experimental settings, thus for the WT system the initial conditions were assumed to reflect that all RII were either free or bound to C with no phosphorylated species or interactions with cAMP. Simulations were then run for the same length

as time as the experiments, assuming the cAMP was added at t = 0 and that autophosphorylation started at that time.

For predicting responses with mutant RIIα subunits, the base model was extended by splitting the RII into two pools, namely RIIα (85%) and RIIβ (15%) but keeping the parameter distribution received from the parameter estimation when only one isoform of RII was accounted for. Experiments with WT RII subunits were successfully re-simulated with the extended model to validate the approach. As the mutations when simulating both S98A and S98E were in the RIIα subtype (85%), the corresponding parameters depicting the mutation were only varied for this pool. Both the mutant forms, S98A and S98E, were tested as different model variants. To mimic the conditions of the S98A mutation in the model, the phosphorylation rates of RIIα and RIIα bound with cAMP were set to zero (i.e. for the RIIα partition of the model, kinetics were restricted to the right-hand square sub-system shown in **Figure 4D**). Here the initial conditions were estimated in the same way as described above. To mimic the S98E mutation in the model, the turnover number for dephosphorylation of pRIIα and pRIIα with cAMP by CN were set to zero (i.e. for the RIIα partition of the model, kinetics were restricted to the left-hand square in **Figure 4D**). Since S98E mimics a case where all the RII subunits are phosphorylated, in this case initial conditions were such that all RIIα were distributed between pRIIα and pRIIα-C.

All model variants were built using the MATLAB Simbiology toolbox (MathWorks). Simulations of these reaction systems were performed using the ode15s solver. All simulations were run for 605 s and the AKAR4 phosphorylation was extracted as output to compare with the experimental findings. The model equations were also exported to the statistical programming language R (https://www.r-project.org/) for implementing the parameter estimation through the ABC-copula approach (**Eriksson et al., 2019**) and only accept parameter sets whose simulated phosphorylated AKAR4 curves reproduced the experimental measurements. A slight modification to the distance measure $\rho$ was introduced to include timeseries data, where $\rho = (\sum_i (y_i^{exp} - y_i^{sim})^2)/n$ where y are experimental and simulated data points (normalized to be within 0 and 1) and $n$ the number of data points for the experiment (for details see the code repository). The sampling resulted in approximately 15,000 parameter samples (a subset of which are shown in **Figure 5—figure supplement 1A**) which all fitted the experimental data within a threshold set to $\rho < 0.01$. All parameter set samples, describing the uncertainty in the parameter estimates, were next projected onto the situations with mutant RIIα subunits. The model immediately reproduced the experimental observations with RIIα S98E subunits. Although the model correctly reproduced lower rates of AKAR4 that occur with RIIα S98A subunits, and that suppression of PKA activity by AKAP79/CN is reduced in this case, there was a substantial spread in the simulated responses in this case. This indicated that WT data had not perfectly constrained the dynamics in the unphosphorylated RII sub-system (right-hand square, **Figure 4B**). Therefore, to better understand which parameter characteristics that were important to also account for the RIIα S98A, the parameter sets were sub-classified based on how well they fit mutation data (**Figure 5B-D**) using a threshold of 0.01. The parameter sets and its effect on different chemical species of the model were described by multi- trajectory, pairwise coordinate and boxplots, where the color schemes follow the classification described above. A code repository for this study may be accessed at https://github.com/jdgas/AKAP79_PKA (copy archived at swh:1:rev:0fb83d341f568ac92340857be7886b3ccc3004b3, **Eriksson, 2021**). It contains the R code for the ABC method as well as MATLAB code for reproducing figures. The R code has to be run on a computer cluster. The repository also contains the models with a few example parameter sets and the full parameter sample as described above.

## Lentivirus Construction

Lentiviruses were generated by inserting RIIα-IRES2-GFP expression cassettes into a pFUGW-H1 lentiviral vector (Addgene cat no. 25870) containing a shRNA sequence targeting for rat RIIα. In the first step, coding sequence for rat PKA RIIα was isolated from a cDNA library that we generated from total hippocampal RNA from a 7 day old male Sprague Dawley rat bred in the UCL colony. RNA was extracted using an RNeasy Mini Kit before the cDNA library was generated using the first-strand cDNA synthesis kit. Coding sequence for RIIα was amplified from the library using primers Prkar2a_F & Prkar2a_R and inserted upstream of the IRES2 sequence in pIRES2-GFP (Clontech) using EcoRI and BamHI entry sites. Three pFUGW-H1-shRIIα vectors were constructed to determine an optimal targeting sequence for knockdown of rat RIIα. The targeting sequences (primer pairs shRIIα_F1/R1, shRIIα_F2/R2, and shRIIα_F3/R3) were inserted using the XbaI site of pFUGW-H1. The efficiency of

each targeting sequence was determined by co-transfecting HEK293T cells with pIRES2-RIIα-EGFP and each pFUGW-H1 vector, with the pFUGW vector in a 10-fold excess. Anti-RIIα immunoblotting revealed that sequence shRIIα–1, which targets bases 134–154 in the rat RIIα coding sequence, was particularly effective at knocking down RIIα protein levels (*Figure 6B*) so this variant served as the parent pFUGW-H1-shRIIα vector in the subsequent steps. The coding sequence for RIIα in pIRES2-RIIα-GFP was rendered shRNA-resistant ('RIIα*') by SDM with primers Prkar2a_shRNA_resist_F & R. After introducing an NheI entry site into pFUGW-H1-shRIIα by SDM using primers FUGW_NheI_F & R, the dual expression cassette for RIIα-IRES2-GFP was transferred across into pFUGW-H1-shRIIα down-stream of the ubiquitin promoter using NheI and AgeI sites to create the complete lentiviral vector pFUGW-H1-shRIIα-RIIα*-IRES2-EGFP.

Vectors containing RIIα replacement sequences with mutations at S97 were obtained by SDM with primers pairs rS97A_F & R and rS97E_F & R. In addition, a control vector containing a scrambled shRNA sequence was constructed using primers shScram_F & R. To produce lentivirus, pFUGW vectors were co-transfected with pCMVdR8.74 packaging vector (Addgene cat no. 12259) and pMD2.G envelope glycoprotein vector (Addgene cat no. 12259) into HEK293 cells using Lipofectamine 2000 and maintained in DMEM supplemented with 10 % FBS. Cell culture media was collected at both 48 and 72 hours after transfection, subjected to 0.45 μm filtering, and centrifuged at 48,384 x *g* for 4 hours at 4 °C to concentrate viral particles. Pelleted virus was resuspended in sterile PBS and stored at – 80 °C. Lentiviruses were validated by transducing dissociated hippocampal cultures on DIV7. Neurons were collected on DIV14, and protein extracted using sonication (3 × 10 s at 20 MHz) in extraction buffer. The homogenate was clarified by centrifugation at 21,130 x g for 15 minutes before analysis of protein levels in the supernatant by immunoblotting using antibodies including anti-PKA pRIIα (Abcam, RRID: AB_779040), anti-GFP (Sigma Aldrich, RRID: AB_2750576), and anti-β-tubulin antibodies (Biolegend, RRID: AB_2565030).

## Lentiviral Infection and Imaging of Dissociated Primary Hippocampal Neurons

Primary hippocampal cultures were cultured from E18 Sprague-Dawley pups. Hippocampi were isolated and triturated with trypsin (0.025%) before plating on poly-L-lysine-coated coverslips or 6-well plates in DMEM containing 10 % heat-inactivated horse serum, and penicillin (40 U/mL)/streptomycin (40 μg/mL). Neurons were cultured at 37 °C in 95 % air/5 % $CO_2$. Two hours after seeding, the plating media was replaced with Neurobasal-A supplemented with 1 % B27, 0.5 % (v/v) GlutaMAX, 20 mM glucose, and penicillin (100 U/mL)/streptomycin (100 μg/mL). Culture media and additives were purchased from Gibco with the exception of GlutaMAX (Thermo Fisher Scientific). Neurons were infected with lentivirus at DIV7 or DIV9 for dendritic spine density and time-lapse experiments, respectively. Concentrated viral stocks were diluted in conditioned media and incubated with neurons for 18 hours before replacing with fresh pre-conditioned media. Live-cell confocal imaging of dendritic spines was performed using an upright Zeiss LSM 510 confocal microscope equipped with an Achroplan 40 x water differential interference contrast objective (numerical aperture 0.8). Trans-duced neurons were washed four times in HEPES-buffered Krebs solution (140 mM NaCl, 4.2 mM KCl, 1.2 mM $MgCl_2$, 2.52 mM $CaCl_2$, 5 mM Na HEPES, and 11 mM glucose, adjusted to pH 7.4 with NaOH) and placed into a chamber in this same solution at room temperature. For each dendritic segment, upper and lower bounds in the z-plane were initially determined using a rapid z-scan. A full image stack was then collected using a 488 nm Argon laser and a 505–530 nm band-pass emission filter for imaging EGFP fluorescence using 512 × 512 frames with 3-line averaging, and optical slice spacing of 1.035 μm. Time-lapse experiments were conducted to measure changes in spine density and spine-head size after the induction of chemical LTD. An optical slice spacing of 0.9 μm was used during time-lapse experiments. Z-stacks were acquired every 5 min from 15 min before to 60 min after the induction of chemical LTD. Bath application of 20 μM NMDA for 3 min was used to induce NMDAR-dependent LTD (*Lee et al., 1998*). Data was deconvolved using ImageJ (NIH) before auto-mated dendrite identification and classification in NeuronStudio (*Rodriguez et al., 2008*). In time-lapse experiments, dendritic spine densities were normalized to the value at t = 0.

## Statistical analysis

All data are presented as means ± SE. Kinetic rates were statistically compared using two-tailed unpaired Student $t$-tests. Spine imaging data was compared by ANOVA with Turkey post-hoc tests (*Figure 6D*) and Bonferroni's post-hoc test (*Figure 6G*). *$P < 0.05$; **$P < 0.01$; ***$P < 0.001$.

## Acknowledgements

We thank Denis Yuan for assistance with protein purification, and Alexandra Jauhiainen, Andrei Kramer and Federica Milinanni for help with the parameter estimation process. MGG is a Wellcome Trust and Royal Society Sir Henry Dale fellow (104194/Z/14 /A), and is grateful for support from the BBSRC (BB/N015274/1). SH is a Rett Syndrome Fellow and also supported by a Wellcome Trust Collaborative award to TGS. The research was supported by the Swedish Research Council (VR-M-2017–02806; VR-M-2020–01652), the Swedish e-Science Research Centre (SeRC), European Union/Horizon 2020 no. 945,539 Human Brain Project SGA3, and an Erasmus Scholarship from Portugal. Optimizations and simulations were performed on resources provided by the Swedish National Infrastructure for Computing (SNIC) at Lunarc, Lund University.

## Additional information

### Funding

| Funder | Grant reference number | Author |
|---|---|---|
| Wellcome Trust | 104194/Z/14/A | Matthew G Gold |
| Royal Society | 104194/Z/14/A | Matthew G Gold |
| Biotechnology and Biological Sciences Research Council | BB/N015274/1 | Matthew G Gold |
| Swedish Research Council | VR-M-2017-02806 | Jeanette Hellgren Kotaleski |
| Horizon 2020 | 945539 Human Brain Project SGA3 | Jeanette Hellgren Kotaleski |
| Erasmus+ | Erasmus Scholarship | João Antunes |
| Wellcome Trust | 217199/Z/19/Z | Saad Hannan Trevor G Smart |
| Swedish Research Council | VR-M-2020-01652 | Jeanette Hellgren Kotaleski |
| Swedish e-Science Research Centre (SeRC) | | Olivia Eriksson Jeanette Hellgren Kotaleski |

The funders had no role in study design, data collection and interpretation, or the decision to submit the work for publication.

### Author contributions

Timothy W Church, Conceptualization, Funding acquisition, Investigation, Supervision, Visualization, Writing – original draft, Writing – review and editing; Parul Tewatia, Investigation, Software, Visualization, Writing – original draft, Writing – review and editing; Saad Hannan, João Antunes, Olivia Eriksson, Investigation, Software, Visualization, Writing – review and editing; Trevor G Smart, Jeanette Hellgren Kotaleski, Funding acquisition, Investigation, Software, Supervision, Visualization, Writing – review and editing; Matthew G Gold, Conceptualization, Funding acquisition, Investigation, Software, Supervision, Visualization, Writing – original draft, Writing – review and editing

### Author ORCIDs

Timothy W Church (iD) http://orcid.org/0000-0002-5958-6304
Parul Tewatia (iD) http://orcid.org/0000-0002-3096-1318

Saad Hannan  http://orcid.org/0000-0003-4594-0808
João Antunes  http://orcid.org/0000-0001-9635-5145
Olivia Eriksson  http://orcid.org/0000-0003-0740-4318
Trevor G Smart  http://orcid.org/0000-0002-9089-5375
Jeanette Hellgren Kotaleski  http://orcid.org/0000-0002-0550-0739
Matthew G Gold  http://orcid.org/0000-0002-1281-0815

### Ethics

Experiments involving rats were done in accordance with the United Kingdom Animals Act, 1986 and within University College London Animal Research guidelines overseen by the UCL Animal Welfare and Ethical Review Body under project code 14058.

### Decision letter and Author response

Decision letter https://doi.org/10.7554/eLife.68164.sa1
Author response https://doi.org/10.7554/eLife.68164.sa2

---

## Additional files

### Supplementary files

• Supplementary file 1. Kinetic modeling parameters.The table lists parameters used in the computational modeling. Parameter terminology is according to the numbers above stated in *Figure 4D*, for example k12 refers to the on rate of cAMP binding to state 1 (pRII-C) to produce state 2 (pRII-C-cAMP). The prior range used to constrain parameter estimation is provided for each parameter along with links to the references used to set the default values.

• Supplementary file 2. Oligonucleotide primer sequences.

• Transparent reporting form

• Source data 1. Original images of Coomassie-stained gels and immunoblots included in the manuscript.

### Data availability

Source data files have been provided for figures 1-6, figure 1-supplement 2, figure 1-supplement 3, figure 3-supplement 1, and figure 3-supplement 2. Original images and uncropped images for Coomassie-stained gels and immunoblots presented in the manuscript are shown in the zipped folder provided as an additional file. A code repository for this study may be accessed at https://github.com/jdgas/AKAP79_PKA, (copy archived swh:1:rev:0fb83d341f568ac92340857be7886b3ccc3004b3). It contains the R code for the ABC method as well as MATLAB code for reproducing figures. The R code has to be run on a computer cluster. The repository also contains the models with a few example parameter sets, and the full parameter sample as described above.

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
