## [Decision Letter]

**Acceptance summary:**

The revised manuscript very nicely addresses the reviewer comments and provides rigorous biochemical data supporting a model for PKA inactivation wherein dephosphorylation of the PKA regulatory subunit within a multiprotein complex leads to rapid capture of the PKA catalytic subunit limiting signaling duration. The findings provide a tantalizing mechanism to selectively modulate PKA activity at precise subcellular locations. This work will be of interest to neuroscientists as well as a broad audience of cell biologists, as it provides new insight into the myriad of cellular functions regulated by the well-studied cAMP-dependent protein kinase, PKA.

**Decision letter after peer review:**

Thank you for submitting your article "AKAP79 enables calcineurin to directly suppress protein kinase A activity" for consideration by *eLife*. Your article has been reviewed by 3 peer reviewers, including Amy Andreotti as Reviewing Editor and Reviewer #1, and the evaluation has been overseen by Philip Cole as the Senior Editor. The following individual involved in review of your submission has agreed to reveal their identity: Susan S Taylor (Reviewer #2).

Recommendations for the authors:

Please revise and respond to reviewer comments with particular focus on strengthening the in vitro findings of this work.

Additional questions are as follows:

What about PDEs? Are they part of this signaling unit in cells? It seems as though they would be essential to terminal the cAMP signal. There is evidence for channeling here where the preferred substrate for PDEs may be cAMP bound to an R subunit.

To further substantiate a physiological role of the mechanism proposed experiments in living cells could eventually be performed where ∆CaN and ∆PKA mutants of AKAP79 are expressed and the level of phosphorylation of the associated RII is measured. In the same conditions, the level of phosphorylation of physiological PKA targets that associate to AKAP79 should also be investigated.

A recent review (Gildart et al., J Physiol 598.14 (2020) pp 3029-3042) briefly mentions the idea that "Inclusion of CaN in AKAP complexes could enhance RII dephosphorylation, thereby facilitating re-formation of PKA holoenzyme and limiting PKA signalling duration" but then immediately states that this mechanism has not been fully tested. The submitted work of Church et al., now test this mechanism in vitro and so the authors might consider including a reference to this review (and possibly additional papers cited therein) in their manuscript.

The importance of the linker region N-terminal to the inhibitor site in the RII subunit for recognition by CN was first described by Blumenthal, et al. (JBC, 1986.261:8140). This reference should be cited in addition to the Stemmer and Klee paper even though both were a long time ago.

---

## [Author Response]

Recommendations for the authors:Please revise and respond to reviewer comments with particular focus on strengthening the in vitro findings of this work.

We have strengthened the in vitro findings of our work by incorporating the following additions. We have assayed CN activity towards both pNPP and pRII phosphopeptide with WT, ΔCN, and ΔPKΔ variants of AKAP79c97. The results of these experiments are shown in shown in the new Figure 1—figure supplement 3, and described in additional text in the Results section (p. 4, lines 130-140). Additional methods for pNPP and pRII phosphopeptide dephosphorylation assays are also provided (p. 21, lines 525-540). We also include new experiments comparing the rate of AKAR4 phosphorylation by PKA C subunits either alone, with CN, or with PP1 (new Figure 3—figure supplement 2A), and have compared rates of pAKAR4 dephosphorylation with different concentrations of CN and PP1 (Figure 3- figure supplement 2B). The results of these experiments are described in the Results section on p. 6, lines 176-182, and additional methods information is included for pAKAR4 dephosphorylation assays and PP1 preparation on p. 23, lines 586-596 and p. 20, lines 483-488, respectively. We have also improved the modelling by initially fitting against experimental data collected at 0, 0.2, 1 and 2 μM cAMP as described in the Results section on p. 7, lines 228-236. Our simulations now reproduce responses of mixtures stimulated with different concentrations of cAMP (Figure 4—figure supplement 2). Updates to the modelling approach are described on p. 24, lines 625-647, p. 25, lines 668-672, and in Supplementary File 1. We have assembled an extensive code repository with instructions for running simulations, which is publicly available at https://github.com/jdgas/AKAP79_PKA and referred to in the manuscript on p. 26, lines 684-689. We have also included several further textual changes, which are described in our other responses.

Additional questions are as follows:What about PDEs? Are they part of this signaling unit in cells? It seems as though they would be essential to terminal the cAMP signal. There is evidence for channeling here where the preferred substrate for PDEs may be cAMP bound to an R subunit.

To our knowledge there is no robust evidence of a direct interaction between a PDE and a member of the AKAP79 signalling complex but clearly PDEs are essential to terminating any cAMP signal. We have included additional text in the discussion referring to PDE channeling (p. 12, lines 398-399), as follows:

“should also be considered along with PDEs that can terminate cAMP signals with high spatiotemporal precision (Tulsian et al., Channeling of cAMP in PDE-PKA Complexes Promotes Signal Adaptation, Biophys J, 2017, PMID 28636912; Bock et al., Cell, 2000)”.

To further substantiate a physiological role of the mechanism proposed experiments in living cells could eventually be performed where ∆CaN and ∆PKA mutants of AKAP79 are expressed and the level of phosphorylation of the associated RII is measured. In the same conditions, the level of phosphorylation of physiological PKA targets that associate to AKAP79 should also be investigated.

We agree with the reviewer that this will be a sensible strategy for future investigations.

We are developing lentiviral vectors for expression of epitope-tagged variants of AKAP150 in rat hippocampal slices with the aim of monitoring RII phosphorylation state in living neurons using immunoblotting as in (Zhang et al., PLoS Biol, 2015, PMID 26158466) and (Isensee et al., J Cell Biol, 2018, PMID 29615473).

A recent review (Gildart et al., J Physiol 598.14 (2020) pp 3029-3042) briefly mentions the idea that "Inclusion of CaN in AKAP complexes could enhance RII dephosphorylation, thereby facilitating re-formation of PKA holoenzyme and limiting PKA signalling duration" but then immediately states that this mechanism has not been fully tested. The submitted work of Church et al., now test this mechanism in vitro and so the authors might consider including a reference to this review (and possibly additional papers cited therein) in their manuscript.

We have included a citation of this work on page 3, line 78-79, with the statement:

“… and therefore theoretically an AKAP might support pRII dephosphorylation by CN in cells (Gildart et al., 2020)”.

The importance of the linker region N-terminal to the inhibitor site in the RII subunit for recognition by CN was first described by Blumenthal, et al. (JBC, 1986.261:8140). This reference should be cited in addition to the Stemmer and Klee paper even though both were a long time ago.

We have included a citation of this work on page 3, line 73, thank you for bringing it to our attention. We thank the reviewers for their insightful and constructive comments.